



**High organic inputs explain shallow and deep SOC storage in a long-term agroforestry**
**system – Combining experimental and modeling approaches.**
Rémi Cardinael[a,b,c*], Bertrand Guenet[d], Tiphaine Chevallier[a], Christian Dupraz[e], Thomas
Cozzi[b], Claire Chenu[b]
[a] IRD, UMR Eco&Sols, Montpellier SupAgro, 2 place Viala, 34060 Montpellier, France
[b] AgroParisTech, UMR Ecosys, Avenue Lucien Brétignières, 78850 Thiverval-Grignon, France
[c] CIRAD, UPR AIDA, Avenue d'Agropolis, 34398 Montpellier, France (present address)
[d] Laboratoire des Sciences du Climat et de l'Environnement, UMR CEA-CNRS-UVSQ, CE
L'Orme des Merisiers, 91191 Gif-Sur-Yvette, France
[e] INRA, UMR System, Montpellier SupAgro, 2 place Viala, 34060 Montpellier, France
* Corresponding author. Tel.: +33 04.67.61.53.08. E-mail address: remi.cardinael@cirad.fr
*Keywords:* priming effect, deep roots, deep soil organic carbon, spatial heterogeneity,
silvoarable system, crop yield, SOC modeling
**Abstract**
Agroforestry is an increasingly popular farming system enabling agricultural diversification
and providing several ecosystem services. In agroforestry systems, soil organic carbon (SOC)
stocks are generally increased, but it is difficult to disentangle the different factors responsible
for this storage. Organic carbon (OC) inputs to the soil may be larger, but SOC decomposition
rates may be modified owing to microclimate, physical protection, or priming effect from roots,
especially at depth. We used an 18-year-old silvoarable system associating hybrid walnut trees
(*Juglans regia × nigra*) and durum wheat (*Triticum turgidum* L. subsp. *durum*), and an adjacent





agricultural control plot to quantify all OC inputs to the soil - leaf litter, tree fine root
senescence, crop residues, and tree row herbaceous vegetation -, and measure SOC stocks down
2 m depth at varying distances from the trees. We then proposed a model that simulates SOC
dynamics in agroforestry accounting for both the whole soil profile and the lateral spatial
heterogeneity.
OC inputs to soil were increased by about 40% (+ 1.11 t C ha$^{-1}$ yr$^{-1}$) down to 2 m depth in the
agroforestry plot compared to the control, resulting in an additional SOC stock of 6.3 t C ha$^{-1}$
down to 1 m depth. The model described properly the measured SOC stocks and distribution
with depth. It showed that the increased inputs of fresh biomass to soil explained the observed
additional SOC storage in the agroforestry plot. Moreover, modeling revealed a strong priming
effect that would reduce the potential SOC storage due to higher organic inputs in the
agroforestry system by 75 to 90%. This result questions the potential of soils to store large
amounts of carbon, especially at depth. Deep-rooted trees modify OC inputs to soil, a process
that deserves further studies given its potential effects on SOC dynamics.
**1 Introduction**
Agroforestry systems are complex agroecosystems combining trees and crops or pastures
within the same field (Nair, 1993, 1985; Somarriba, 1992). More precisely, silvoarable systems
associate parallel tree rows with annual crops. Some studies showed that these systems could
be very productive, with a land equivalent ratio (Mead and Willey, 1980) reaching up to 1.3
(Graves et al., 2007). Silvoarable systems may therefore produce up to 30% more marketable
biomass on the same area of land compared to crops and trees grown separately. This
performance can be explained by a better use of water, nutrients and light by the agroecosystem
throughout the year. Trees grown in silvoarable systems usually grow faster than the same trees
grown in forest ecosystems, because of their lower density, and because they also benefit from



the crop fertilization (Balandier and Dupraz, 1999; Chaudhry et al., 2003; Chifflot et al., 2006).
In temperate regions, farmers usually grow one crop per year, and this association of trees can
extend the growing period at the field scale, especially when winter crops are intercropped with
trees having a late bud break (Burgess et al., 2004). However, after several years, a decrease of
crop yield can be observed in mature and highly dense plantations, especially close to the trees,
due to competition between crops and trees for light, water, and nutrients (Burgess et al., 2004;
Dufour et al., 2013; Yin and He, 1997).
Part of the additional biomass produced in agroforestry is used for economical purposes, such
as timber or fruit production. Leaves, tree fine roots, pruning residues and the herbaceous
vegetation growing in the tree rows will usually return to the soil, contributing to a higher input
of organic carbon (OC) to the soil compared to an agricultural field (Peichl et al., 2006).
In such systems, the observed soil organic carbon (SOC) stocks are also generally higher
compared to a cropland (Albrecht and Kandji, 2003; Kim et al., 2016; Lorenz and Lal, 2014).
Cardinael *et al.*, (2017) measured a mean SOC stock accumulation rate of 0.24 (0.09-0.46) t C
ha$^{-1}$ yr$^{-1}$ at 0-30 cm depth in several silvoarable systems compared to agricultural plots in
France. Higher SOC stocks were also found in Canadian agroforestry systems, but measured
only to 20 cm depth (Bambrick et al., 2010; Oelbermann et al., 2004; Peichl et al., 2006).
To our knowledge, we are still not able to disentangle the factors responsible for such a higher
SOC storage. This SOC storage might be due to higher OC inputs but it could also be favored
by a modification of the SOC decomposition owing to a change in SOC physical protection
(Haile et al., 2010), and/or in soil temperature and moisture.
The introduction of trees in an agricultural field modifies the amount, but also the distribution
of fresh organic carbon (FOC) input to the soil, both vertically and horizontally (Bambrick et
al., 2010; Howlett et al., 2011; Peichl et al., 2006). FOC inputs from the trees decrease with
increasing distance from the trunk and with soil depth (Moreno et al., 2005). On the contrary,





crop yield usually increases with increasing distance from the trees (Dufour et al., 2013; Li et
al., 2008). Therefore, the proportions of FOC coming from both the crop residues and the trees
change with distance from the trees, soil depth, and time.
Tree fine roots (diameter ≤ 2 mm) are the most active part of root systems (Eissenstat and Yanai,
1997) and play a major role in carbon cycling. In silvoarable systems, tree fine root distribution
within the soil profile is strongly modified due to the competition with the crop, inducing a
deeper rooting compared to trees grown in forest ecosystems (Cardinael et al., 2015a; Mulia
and Dupraz, 2006). Deep soil layers may therefore receive significant OC inputs from fine root
mortality and exudates. Root carbon has a higher mean residence time in the soil compared to
shoot carbon (Kätterer et al., 2011; Rasse et al., 2006), presumably because root residues are
preferentially stabilized within microaggregates or adsorbed to clay particles. Moreover,
temperature and moisture conditions are more buffered in the subsoil than in the topsoil. The
microbial biomass is also smaller at depth (Eilers et al., 2012; Fierer et al., 2003), and the spatial
segregation with organic matter is larger (Salomé et al., 2010) resulting in lower decomposition
rates. Deep root carbon input in the soil could therefore contribute to a SOC storage with high
mean residence times. However, some studies showed that adding FOC – a source of energy
for microorganisms - to the subsoil enhanced decomposition of stabilized carbon, a process
called « priming effect »  (Fontaine et al., 2007). The priming effect is stronger when induced
by labile molecules like root exudates than by root litter coming from the decomposition of
dead roots (Shahzad et al., 2015). Therefore, the net effect of deep roots on SOC stocks has to
be assessed, especially in silvoarable systems.
Models are crucial as they allow virtual experiments to best design and understand complex
processes in these systems (Luedeling et al., 2016). Several models have been developed to
simulate interactions for light, water and nutrients between trees and crops (Charbonnier et al.,
2013; Duursma and Medlyn, 2012; van Noordwijk and Lusiana, 1999; Talbot, 2011) or to



predict tree growth and crop yield in agroforestry systems (Graves et al., 2010; van der Werf et
al., 2007). However, none of these models are designed to simulate SOC dynamics in
agroforestry systems and they are therefore not useful to estimate SOC storage. Oelbermann &
Voroney (2011) evaluated the ability of the CENTURY model (Parton et al., 1987) to predict
SOC stocks in tropical and temperate agroforestry systems, but with a single-layer modeling
approach (0-20 cm). The approach of modeling a single topsoil layer assumes that deep SOC
does not play an active role in carbon cycling, while it was shown that deep soil layers contain
important amounts of SOC (Jobbagy and Jackson, 2000), and that part of this deep SOC could
cycle on decadal timescales due to root inputs or to dissolved organic carbon transport (Baisden
and Parfitt, 2007; Koarashi et al., 2012). The need to take into account deep soil layers when
modeling SOC dynamics is now well recognized in the scientific community (Baisden et al.,
2002; Elzein and Balesdent, 1995), and several models have been proposed (Ahrens et al., 2015;
Braakhekke et al., 2011; Guenet et al., 2013; Koven et al., 2013; Taghizadeh-Toosi et al., 2014).
Using vertically discretized soils is particularly important when modeling the impact of
agroforestry systems on SOC stocks, but to our knowledge, vertically spatialized SOC models
have not yet been tested for these systems.

The aims of this study were then twofold: (i) to propose a model of soil C dynamics in
agroforestry systems able to account for both vertical and lateral spatial heterogeneities and (ii)
to test whether variations of fresh organic carbon (FOC) input could explain increased SOC
stocks both using experimental data and model runs.
For this, we first compiled data on FOC inputs to the soil obtained in a 18-year-old agroforestry
plot and in an agricultural control plot in southern France, in which SOC stocks have been
recently quantified to 2 m depth (Cardinael et al., 2015b). FOC inputs comprised tree fine roots,
tree leaf litter, aboveground and belowground biomass of the crop and of the herbaceous



vegetation in the tree rows. We compiled recently published data for FOC inputs (Cardinael et
al., 2015a; Germon et al., 2016), and measured the others (Table 1).

We then modified a two pools model proposed by Guenet *et al.*, (2013), to create a spatialized
model over depth and distance from the tree, the CARBOSAF model (soil organic CARBOn
dynamics in Silvoarable AgroForestry systems). Based on data acquired since the tree planting
in 1995 (crop yield, tree growth), and on FOC inputs, we modeled SOC dynamics to 2 m depth
in both the silvoarable and agricultural control plot. We evaluated the model against measured
SOC stocks along the profile and used this opportunity to test the importance of priming effect
(*PE*) for deep soil C dynamics in a silvoarable system. The performance of the two pools model
including *PE* was also compared with a model version including three OC pools.

**2 Materials and methods**
**2.1 Study site**
The experimental site is located at the Restinclières farm Estate in Prades-le-Lez, 15 km North
of Montpellier, France (longitude 04°01' E, latitude 43°43' N, elevation 54 m a.s.l.). The
climate is sub-humid Mediterranean with an average temperature of 15.4°C and an average
annual rainfall of 973 mm (years 1995–2013). The soil is a silty and carbonated (pH = 8.2) deep
alluvial Fluvisol (IUSS Working Group WRB, 2007). In February 1995, a 4.6 hectare
silvoarable agroforestry plot was established with the planting of hybrid walnut trees (*Juglans*
*regia* × *nigra* cv. NG23) at a density of 192 trees ha[-1] but later thinned to 110 trees ha[-1]. Trees
were planted at 13 m × 4 m spacing, and tree rows are East–West oriented. The cultivated alleys
are 11 m wide. The remaining part of the plot (1.4 ha) was kept as an agricultural control plot.
Since the tree planting, the agroforestry alleys and the control plot were managed in the same
way. The associated crop is most of the time durum wheat (*Triticum turgidum* L. subsp. *durum*),



except in 1998, 2001 and 2006, when rapeseed (*Brassica napus* L.) was cultivated, and in 2010
and 2013, when pea (*Pisum sativum* L.) was cultivated. The soil is ploughed to a depth of 0.2
m before sowing, and the wheat crop is fertilized with an average of 120 kg N ha$^{-1}$ yr$^{-1}$. Crop
residues (wheat straw) are also exported, but about 25% remain on the soil. Tree rows are
covered by spontaneous herbaceous vegetation. Two successive herbaceous vegetation types
occur during the year, one in summer and one in winter. The summer vegetation is mainly
composed of *Avena fatua* L., and is 1.5 m tall. In winter, the vegetation is a mix of *Achillea*
*millefolium* L., *Galium aparine* L., *Vicia* L., *Ornithogalum umbellatum* L. and *Avena fatua* L,
and is 0.2 m tall.

**Table 1.** Synthesis of the different field and laboratory data available or measured, and their

162         sources.

| Description of the data | Source |
|---|---|
| Soil texture, bulk densities, SOC stocks | Cardinael *et al.*, (2015a) |
| Soil temperature and soil moisture | Measured |
| Tree growth (DBH) | Measured |
| Tree wood density | (Talbot, 2011) |
| Tree fine root biomass | Cardinael *et al.*, (2015b) |
| Tree fine root turnover | Germon *et al.*, (2016) |
| Crop yield and crop ABG biomass | Dufour *et al.*, (2013) and measured |
| Crop root biomass | Prieto *et al.*, (2015) and measured |
| Tree row herbaceous vegetation – ABG biomass | Measured |
| Tree row herbaceous vegetation – root biomass | Measured |
| Biomass carbon concentrations | Measured |
| Potential decomposition rate of roots | Prieto *et al.*, (2016a) |
| HSOC potential decomposition rate | Measured |

DBH: Diameter at Breast Height; ABG: aboveground; OC: organic carbon; HSOC: humified
soil organic carbon.

**2.2 Organic carbon stocks**
**2.2.1 Soil organic carbon stocks**



SOC data have been published in Cardinael *et al.*, (2015a). Briefly, soil cores were sampled
down to 2 m depth in May 2013, 100 in the agroforestry plot, and 93 in the agricultural control
plot. SOC concentrations, SOC stocks, and soil texture were measured for ten soil layers (0.0-
0.1, 0.1-0.3, 0.3-0.5, 0.5-0.7, 0.7-1.0, 1.0-1.2, 1.2-1.4, 1.4-1.6, 1.6-1.8, and 1.8-2.0 m). In the
agroforestry plot, 40 soil cores were taken in the tree rows, while 60 were sampled in the alleys
at varying distances from the trees. Soil organic carbon stocks were quantified on an equivalent
soil mass basis (Ellert and Bettany, 1995).

**2.2.2 Tree aboveground and stump carbon stocks**
Three hybrid walnuts were chopped down in 2012. The trunk circumference was measured
every meter up to the maximum height of the tree to estimate its volume. The trunk biomass
was estimated by multiplying the trunk volume by the wood density that was measured at 616
kg m$^{-3}$ during a previous work at the same site (Talbot, 2011). Then, branches were cut, the
stump was uprooted, and they were weighted separately. Samples were brought to the
laboratory to determine the moisture content, which enabled calculation of the branches and the
stump dry mass.

**2.3 Measurements of organic carbon inputs in the field**
**2.3.1 Carbon inputs from tree fine root mortality**
The tree fine root (diameter ≤ 2 mm) biomass was quantified and coupled with an estimate of
the tree fine root turnover in order to predict the carbon input to the soil from the tree fine root
mortality. A detailed description of the methods used to estimate the tree fine root biomass can
be found in Cardinael *et al.*, (2015b). In March 2012, a 5 (length) × 1.5 (width) × 4 m (depth)
pit was open in the agroforestry plot, perpendicular to the tree row, at the North of the trees.
The tree fine root distribution was mapped down 4 m depth, and the tree fine root biomass was



quantified in the tree row and in the alley. Only results concerning the first two meters of soil,
among those obtained by Cardinael *et al.*, (2015b) will be presented here.
In July 2012, sixteen minirhizotrons were installed in the agroforestry pit, at 0, 1, 2.5 and 4 m
depth, and at two and five meters from the trees. The tree root growth and mortality was
monitored during one year using a scanner (CI-600 Root Growth Monitoring System, CID,
USA), and analyzed using the WinRHIZO Tron software (Régent, Canada). A detailed
description of the methods and of results used to estimate the tree fine root turnover can be
found in Germon *et al.*, (2016).

**2.3.2 Tree litterfall**
In 2009, the crowns of two walnut trees were packed with a net in order to collect the leaf
biomass from September to January. The same was done in 2012 with three other walnut trees.
The leaf litter was then dried, weighted and analyzed for C to quantify the leaf carbon input per
tree.

**2.3.3 Aboveground and belowground input from the crop**
Since the tree planting in 1995, the crop yield was measured 14 times (in 1995, 2000, 2002,
2003, 2004, 2005, 2007, 2008, 2009, 2010, 2011, 2012, 2013, and 2014), while the wheat straw
biomass and the total aboveground biomass were measured six times (in 2007, 2008, 2009,
2011, 2012, and 2014) in both the control and the agroforestry plot (Dufour et al., 2013), using
sampling subplots of 1 m$^2$ each. In the control plot, five subplots have been sampled while in
the agroforestry plot five transects have been sampled. Each transect was made of three
subplots, 2 m North from the tree, 2 m South from the tree, and 6.5 m from the tree (middle of
the alley). In March 2012, a 2 m deep pit was opened in the agricultural control plot (Prieto et
al., 2015), and the root biomass was quantified to the maximum rooting depth (1.5 m). The



root:shoot ratio of durum wheat was measured in the control plot. We assumed that the crop
root biomass turns out once a year, after the crop harvest.

**2.3.4 Above and belowground input from the tree row herbaceous vegetation**
As two types of herbaceous vegetation grow in the tree rows during the year, samples were
taken in summer and winter. In late June 2014, twelve subplots of 1 m$^2$ each were positioned
in the tree rows, around 4 walnut trees. In January 2015, six subplots of 1 m$^2$ each were
positioned in the tree rows, around 2 walnut trees. The middle of each subplot was located at 1
m, 2 m and 3 m, respectively, from the selected walnut tree. All the aboveground vegetation
was collected in each square. In the middle of each subplot, root biomass was sampled with a
cylindrical soil corer (inner diameter of 8 cm). Soil was taken at three soil layers, 0.0-0.1, 0.1-
0.3 and 0.3-0.5 m. In the laboratory, soil was gently washed with water through a 2 mm mesh
sieve, and roots were collected. Roots from the herbaceous vegetation were easily separated
manually from walnut roots, as they were soft and yellow compared to walnuts roots that were
black. After being sorted out from the soil and cleaned, the root biomass was dried at 40°C and
measured.

**2.4 Carbon concentration measurements**
All organic carbon measurements were performed with a CHN elemental analyzer (Carlo Erba
NA 2000, Milan, Italy), after samples were oven-dried at 40°C for 48 hours (Table 2). Dry
biomasses (t DM ha$^{-1}$) of each organic matter inputs were multiplied by their respective organic
carbon concentrations (mg C g$^{-1}$) to calculate organic carbon stocks (t C ha$^{-1}$).

**Table 2.** Organic carbon concentrations and C:N ratio of the different types of biomass.

| Type of biomass | Organic C concentration (mg C g$^{-1}$) | C:N | Number of replicates |
|---|---|---|---|



| | | | |
|---|---|---|---|
| Walnut trunk | $445.7 \pm 1.0$ | $159.1 \pm 25.2$ | 3 |
| Walnut branches | $428.6 \pm 1.7$ | $62.2 \pm 11.7$ | 3 |
| Wheat straw | $433.2 \pm 0.7$ | $55.5 \pm 2.1$ | 5 |
| Wheat root | $351.4 \pm 19$ | $24.8 \pm 2.1$ | 8 |
| Walnut leaf | $449.4 \pm 3.7$ | $49.1 \pm 0.4$ | 3 |
| Walnut fine root | $437.0 \pm 3.3$ | $28.6 \pm 3.4$ | 8 |
| Summer vegetation (ABG) | $448.4 \pm 1.9$ | $37.8 \pm 2.2$ | 5 |
| Summer vegetation (roots) | $314.5 \pm 8.3$ | $33.8 \pm 1.7$ | 6 |
| Winter vegetation (ABG) | $447.7 \pm 5.3$ | $11.2 \pm 0.4$ | 3 |
| Winter vegetation (roots) | $397.4 \pm 5.0$ | $24.7 \pm 0.7$ | 3 |

The organic matter called "vegetation" stands for the herbaceous vegetation that grows in the
tree row. ABG: aboveground. Errors represent standard errors.

**2.5 General description of the CARBOSAF model**
**2.5.1 Organic carbon decomposition**
We adapted a model developed by Guenet et al. (2013) where total SOC is split in two pools,
the FOC and the humified soil organic carbon (HSOC) for each soil layer (Fig. 1a). Input to the
FOC pool comes from the plant litter and the distribution of this input within the profile is
assumed to depend upon depth from the surface ($z$), distance from the tree ($d$), and time ($t$).
Equations describing inputs to the FOC pool ($I_{t,z,d}$) at a given time, depth, and distance are
fully explained in the Results.

The FOC mineralisation is assumed to be governed by first order kinetics, being proportional
to the FOC pool, as given by:
$$\frac{\partial FOC_{t,z,d}}{\partial t} = -k_{FOC} \times FOC_{t,z,d} \times f_{clay,z} \times f_{moist,z} \times f_{temp,z} \qquad (1)$$
where $FOC_{t,z,d}$ is the FOC carbon pool (kg C m$^{-2}$) at a given time ($t$, in years), depth ($z$, in m)
and distance ($d$, in m), and $k_{FOC}$ is its decomposition rate. The potential decomposition rates of
the different plant materials were assessed with a 16-week incubation experiment during a
companion study at the site (Prieto et al., 2016). The decomposition rate $k_{FOC}$ was weighted by
the respective contribution of each type of plant litter as a function of the tree age, soil depth





and distance from the tree. The rate modifiers $f_{clay,z}$, $f_{moist,z}$ and $f_{temp,z}$ are functions depending
respectively on the clay content, soil moisture and soil temperature at a given depth $z$, and range
between 0 and 1.

The $f_{clay}$ function originated from the CENTURY model (Parton et al., 1987):
$$f_{clay,z} = 1 - 0.75 \times Clay_z \qquad (2)$$
where $Clay_z$ is the clay fraction (ranging between 0 and 1) of the soil at a given depth $z$.


**Fig. 1.** Schematic representation of the pools and the fluxes of the (a) two pools model and (b)

272        three pools model.




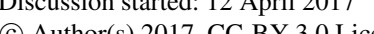


The $f_{moist,z}$ function originated from the meta-analysis of Moyano *et al.*, (2012) and is affected
by soil properties (clay content, SOC content). Briefly, the authors fitted linear models on 310
soil incubations to describe the effect of soil moisture on decomposition. Then, they normalized
such linear models between 0 and 1 to apply these functions to classical first order kinetics. All
details are described in Moyano *et al.*, (2012). To save computing time, we calculated $f_{moist,z}$
only once using measured SOC stocks instead of using modelled SOC stocks and repeated the
calculation at each time step.

The temperature sensitivity of the soil respiration is expressed as $Q_{10}$:
$$f_{temp,z} = Q_{10}^{\frac{temp_z - temp_{opt}}{10}} \qquad (3)$$

with $temp_z$ being the soil temperature in K at each soil depth $z$ and $temp_{opt}$ a parameter fixed to
304.15 K. The $Q_{10}$ value was fixed to 2, a classical value used in models (Davidson and
Janssens, 2006).

Once the FOC is decomposed, a fraction is humified ($h$) and another is respired as $CO_2$ ($1-h$)
(Fig. 1a) following equations (4) and (5).
$$Humified\ FOC_{t,z,d} = h \times \frac{\partial FOC_{t,z,d}}{\partial t} \qquad (4)$$

$$Respired\ FOC_{t,z,d} = (1-h) \times \frac{\partial FOC_{t,z,d}}{\partial t} \qquad (5)$$


Two mathematical approaches are available in the model to describe the mineralisation of
HSOC: a first order kinetics, as given by Eq. (6) or an approach developed by Wutzler &
Reichstein, (2008) and by Guenet *et al.*, (2013) introducing the priming effect, i.e., the
mineralisation of HSOC depends on FOC availability, and given by Eq. (7):
$$\frac{\partial HSOC_{t,z,d}}{\partial t} = -k_{HSOC,z} \times HSOC_{t,z,d} \times f_{moist,z} \times f_{temp,z} \qquad (6)$$



$$\frac{\partial HSOC_{t,z,d}}{\partial t} = -k_{HSOC,z} \times HSOC_{t,z,d} \times (1 - e^{-PE \times FOC_{t,z,d}}) \times f_{moist,z} \times f_{temp,z} \quad (7)$$
where $HSOC_{t,z,d}$ is the humified SOC carbon pool at a given time ($t$, in years), depth ($z$, in m)
and distance ($d$, in m), $k_{HSOC,z}$ is its decomposition rate (yr$^{-1}$) at a given depth $z$, and $PE$ is the
priming effect parameter. The parameters $f_{moist,z}$ and $f_{temp,z}$ are functions depending respectively
on soil moisture and soil temperature at a given depth $z$, and affecting the decomposition rate
of HSOC. They correspond to the moisture equation from Moyano *et al.*, (2012) and to Eq. (3),
respectively. The decomposition rate $k_{HSOC,z}$ was an exponential law depending on soil depth
($z$) as shown by an incubation study (see paragraph *HSOC decomposition rate* further in the
M&M):
$$k_{HSOC,z} = a \times e^{-b \times z} \quad (8)$$
The $b$ parameter of this equation represented the ratio of labile C/stable C within the HSOC
pool. The effect of clay on HSOC decomposition was implicitly taken into account in this
equation as clay content increased with soil depth.
A fraction of decomposed HSOC returns to the FOC assuming that part of the HSOC
decomposition products is as labile as FOC ($h$) and another is respired as $CO_2$ (Fig. 1a) in the
two pools model.

Finally, we also developed an alternative version of the model with three pools by splitting the
HSOC pools into two pools with different turnover rates, HSOC2 being more stabilized than
HSOC1 (Fig. 1b). The non-respired decomposed FOC is split between HSOC1 and HSOC2
following a parameter $f_1$. The non-respired decomposed HSOC1 is split between HSOC2 and
FOC following a parameter $f_2$ whereas non-respired decomposed HSOC2 is only redistributed
into the FOC pools. The decomposition of HSOC1 and HSOC2 both follow the equation (8)
but with different parameter values for $a$.



### 2.5.2 Carbon transport mechanisms

The transport of C between the different soil layers was represented by both advection and diffusion mechanisms (Elzein and Balesdent, 1995), which have been shown to usually describe well the C transport in soils (Bruun et al., 2007; Guenet et al., 2013). The advection represents the C transport due to the water infiltration in the soil, while the diffusion represents the C transport due to the fauna activity. The same transport coefficients were applied to the two C pools, FOC and HSOC.

The advection is defined by:

$$F_A = A \times C \qquad (9)$$

where $F_A$ is the flux of $C$ transported downwards by advection, and $A$ is the advection rate (mm yr$^{-1}$).

The diffusion is represented by the Fick's law:

$$F_D = -D \times \frac{\partial^2 C}{\partial z^2} \qquad (10)$$

where $F_D$ is the flux of C transported downwards by diffusion, $-D$ the diffusion coefficient (cm$^2$ yr$^{-1}$) and $C$ the amount of carbon in the pool subject to transport (FOC or HSOC).

To represent the effect of soil tillage ($z \leq 0.2$ m), we added another diffusion term using the Fick's law but with a value of $D$ several orders of magnitude higher to represent the mixing due to tillage. It must be noted that no tillage effect on the decomposition was represented here because of the large unknowns on these aspects (Dimassi et al., 2013; Virto et al., 2012).

In this model, the flux of $C$ transported downwards by the advection and diffusion ($F_{AD}$) was represented as the sum of both mechanisms, following Elzein & Balesdent (1995):





$$F_{AD} = F_A + F_D \qquad (11)$$


The FOC and HSOC pools dynamics in the two pools model correspond to:
$$\frac{\partial FOC}{\partial t} = I_{t,z,d} + \frac{\partial F_{AD}}{\partial z} + h \times \frac{\partial HSOC_{t,z,d}}{\partial t} - \frac{\partial FOC_{t,z,d}}{\partial t} \qquad (12)$$

$$\frac{\partial HSOC}{\partial t} = \frac{\partial F_{AD}}{\partial z} + h \times \frac{\partial FOC_{t,z,d}}{\partial t} - \frac{\partial HSOC_{t,z,d}}{\partial t} \qquad (13)$$


Finally, the FOC, HSOC1 and HSOC2 pools dynamics in the three pools model correspond to:
$$\frac{\partial FOC}{\partial t} = I_{t,z,d} + \frac{\partial F_{AD}}{\partial z} + h \times f_2 \times \frac{\partial HSOC1_{t,z,d}}{\partial t} + h \times \frac{\partial HSOC2_{t,z,d}}{\partial t} - \frac{\partial FOC_{t,z,d}}{\partial t} \qquad (14)$$

$$\frac{\partial HSOC1}{\partial t \partial} = \frac{\partial F_{AD}}{\partial z} + h \times f_1 \times \frac{\partial FOC_{t,z,d}}{\partial t} - \frac{\partial HSOC1_{t,z,d}}{\partial t} \qquad (15)$$

$$\frac{\partial HSOC2}{\partial t} = \frac{\partial F_{AD}}{\partial z} + h \times (1 - f_1) \times \frac{\partial FOC_{t,z,d}}{\partial t} + h \times (1 - f_2) \times \frac{\partial HSOC1_{t,z,d}}{\partial t}$$

$$- \frac{\partial HSOC2_{t,z,d}}{\partial t} \qquad (16)$$


**2.5.3 Depth dependence of HSOC potential decomposition rates**
The shape of the function (i.e. the *b* parameter) describing the HSOC potential decomposition
rate (Eq. (8)) was determined by incubating soils from the control, the alley and the tree row,
and from different soil layers (0.0-0.1, 0.1-0.3, 0.7-1.0 and 1.6-1.8 m). Soils were sieved at 5
mm, and incubated during 44 days at 20°C at a water potential of -0.03 MPa. Evolved $CO_2$ was
measured using a micro-GC at 1, 3, 7, 14, 21, 28, 35, 44 days. The three first measurement
dates corresponded to a pre-incubation period and were not included in the analysis. For a given
depth, the cumulative mineralised SOC was expressed as a percentage of total SOC and was
plotted against the incubation time. The slopes represented the potential SOC mineralisation
rate at a given soil depth and location. The potential SOC mineralisation rates were then plotted



against soil depth (Fig. S1). We used the soil incubations to determine only the *b* parameter of
the curve: with such short term incubations, the SOC decomposition rate over the soil profile
is overestimated because the $CO_2$ measured during the incubations mainly originates from the
labile C pool. The *a* parameter was optimized following the procedure described further.

**2.6 Boundary conditions of the CARBOSAF model**
**2.6.1 Annual aggregates of soil temperature and soil moisture**
In April 2013, eight soil temperature and moisture sensors (Campbell CS 616 and Campbell
107, respectively) were installed in the agroforestry plot at 0.3, 1.3, 2.8 and 4.0 m depth, and at
2 and 5 m from the trees. Soil temperature and moisture were measured for 11 months.
The mean annual soil temperature in the agroforestry plot was described by the following
equation:
$$T = -0.89 \times z + 288.24 \quad (R^2 = 0.99) \quad (17)$$
where *T* is the soil temperature (K) and *z* is the soil depth (m).

The mean annual soil moisture was described with the following equation:
$$\theta = 0.05 \times z + 0.28 \quad (R^2 = 0.99) \quad (18)$$
where $\theta$ is the soil volumetric moisture (cm cm$^{-3}$) and *z* is the soil depth (m).
Due to a lack of data in the agricultural plot, we assumed that the soil temperature and the soil
moisture were the same in the agroforestry tree rows, alleys and in the control plot, but we
further performed a sensitivity analysis of the model on these two parameters.

**2.6.2 Interpolation of tree growth**
The tree growth has been measured in the field since the establishment of the experiment. We
used the diameter at breast height (*DBH*) as a surrogate of the tree growth preferentially to the





tree height as the field measurements were more accurate. Indeed, *DBH* is easier to measure
than height, especially when trees are getting older. To describe the temporal dynamic of *DBH*
since the tree planting, a linear equation was fitted on the data.

**2.6.3 Change of tree litterfall over time**
For the five walnut trees where the leaf biomass was quantified, *DBH* was also measured. The
ratio between the leaf biomass and *DBH* was then calculated for the five replicates. A linear
relationship between the leaf biomass and *DBH* was then considered to describe the increase of
the leaf litter C input with the tree growth.

**2.6.4 Tree fine root C input from mortality**
A decreasing exponential function was fitted on the root biomass data obtained from the pit in
2012 to describe total fine root biomass (*TFRB*) down to 2 m depth as a function of distance
from the tree. We considered a linear increase of *TFRB* with increasing *DBH*, and a linear
regression was performed between *TFRB* in 2012 and *TFRB* in 1996, the first year after planting
(biomass considered as negligible). A changing distribution of tree fine roots within the soil
profile was taken into account with increasing distance to the tree. For this purpose, exponential
functions ($a \times e^{-b \times z}$) were fitted in the alley every 0.5 m distance, and a linear regression
was fitted between their coefficients *a* and *b* and distance from the tree. However, the
distribution of *TFRB* within the soil profile and with the distance to the tree was considered
constant with time. To finally estimate the tree fine root input due to the mortality, *TFRB* was
multiplied by the measured root turnover.

**2.6.5 Aboveground and belowground input from the crop**



As there were more crop yield measurements than straw biomass measurements, the effect of
agroforestry on the crop yield with time was used as an estimate for change in the aboveground
and belowground wheat biomass.
For this, the relative yield ($Rel\ Y_{AF}$) in the agroforestry system was calculated for each year as
the ratio between the agroforestry yield and the control yield. A linear regression was then fitted
between the relative yield and the $DBH$. The variation of crop yield with distance from the trees
was described with a quadratic equation. But as we aimed to predict SOC stocks up to 6.5 m
distance from the trees (middle of the alley), a linear increase of crop yield with increasing
distance from the tree gave similar results as the quadratic equation over the 6.5 m distance and
was more parsimonious. Finally, the ratio between the straw biomass and the crop yield was
calculated as the average of the six measurements, and was considered constant with time. This
ratio was used to convert crop yield into straw biomass.
To estimate fine root biomass of the crop, we hypothesized that the root:shoot ratio of the durum
wheat was the same in both the agroforestry and agricultural plot, in the absence of any
published data on the matter. The wheat root distribution within the soil profile as a function of
total wheat root biomass was described by an exponential fit. Since the same maximum rooting
depth of the crop was observed in the agroforestry plot and in the control plot, we inferred that
the wheat root distribution within the soil profile was not modified by agroforestry, but only its
biomass.

**2.6.6 Aboveground and belowground input from herbaceous vegetation in the tree rows**
We fitted an exponential function to describe the herbaceous root biomass with depth. We
assumed for simplification that the aboveground and belowground biomasses of the herbaceous
vegetation in the tree row were constant over time.



**2.7 Optimization procedure**
Five parameters were optimized with a Bayesian statistical method (Santaren et al., 2007;
Tarantola, 1987, 2005). These parameters were $A$, the advection rate, $D$, the diffusion
coefficient, $h$ the humification yield, $a$ the coefficient of the $k_{HSOC}$ rate from Eq. (10), and $PE$
the priming coefficient. The model was fitted to the SOC stocks data using a Bayesian curve
fitting method described in Tarantola (1987), after a conversion from SOC stocks in kg C m$^{-2}$
to SOC stocks in kg m$^{-3}$ due to the different soil layers' thickness. We aimed to find a parameter
set that minimizes the distance between model outputs and the corresponding observations,
considering model and data uncertainties, and prior information on parameters. With the
assumption of Gaussian errors for both the observations and the prior parameters, the optimal
parameter set corresponds to the minimum of the cost function $J(x)$:
$$J(x) = 0.5 \times \left[ (y - H(x))^t \times R^{-1} \times (y - H(x)) + (x - x_b)^t \times P_b^{-1} \times (x - x_b) \right] \quad (19)$$
that contains both the mismatch between modelled and observed SOC stock and the mismatch
between a priori and optimized parameters. $x$ is the vector of unknown parameters, $x_b$ the
vector of a priori parameter values, $H()$ the model and $y$ the vector of observations. The
covariance matrices $P_b$ and $R$ describe a priori uncertainties on parameters, and observations,
respectively. Both matrices are diagonal as we suppose the observation uncertainties and the
parameter uncertainties to be independent. To determine an optimal set of parameters which
minimizes $J(x)$, we used the BGFS gradient-based algorithm (Tarantola, 1987). We performed
30 optimizations starting with different parameter prior values to check that the results did not
correspond to a local minimum. To optimize the parameters we only used the data coming from
the control plot.

**2.8 Comparison of models**




Model predictions with and without priming effect were compared calculating the coefficients
of determination, root mean square errors (RMSE) and Bayesian information criteria (BIC).
$$RMSE = \sqrt{\frac{1}{N}\sum_{i=1}^{N}(x_i - \bar{x})^2} \qquad (20)$$

where $i$ is the number of observations (1 to N), $x_i$ is the predicted value and $\bar{x}$ is the mean
observed value.
$$BIC = N \times \ln(MSD) + k \times \ln(N) \qquad (21)$$

where $N$ is the number of observations, $MSD$ is the mean squared deviation, and $k$ is the number
of model parameters.

The model was run at a yearly time step using mean annual soil temperature and moisture and
annual C inputs to the soil. SOC pools were initialized after a spin-up of 5000 years in the
control plot. Measured SOC stocks in 2013 in the control plot were used for the spin up. The
associated uncertainty was estimated with the 93 soil cores sampled in the control plot (see
section 2.2.1). Due to a lack of relevant data, we assumed that the climate and the land use were
the same for the last 5000 years, and that SOC stocks in the control plot were at equilibrium.
Therefore, SOC stocks at the end of the spin-up equaled SOC stocks in the control plot. Three
different spin-ups were performed, corresponding to the three different models that were used:
one spin-up with the two pools model without the priming effect, one spin-up with the two
pools model with the priming effect, and one spin-up with the three pools model. In the
agroforestry, the model was run from the ground (0 m) to 2 m depth, and from the tree (0 m) to
6.5 m from the tree (middle of the alley). The model was applied separately across locations of
a tree-distance gradient having varying OC inputs, each soil column was considered
independent from another. The model was then run from $t_0$ to $t_{18}$ (years) after tree planting. The





spatial resolution was 0.1 m both vertically and horizontally. The model was developed using
R 3.1.1 (R Development Core Team, 2013).

**2.9 Estimation of the priming intensity and its impact on SOC storage**
In equation (7), the priming effect (*PE*) is considered as a control of the FOC on the HSOC
decomposition and not as an accelerating factor of the HSOC decomposition. This method
followed the Wutzler & Reichstein, (2008) approach based on the microbial biomass and
adapted to the FOC by Guenet *et al.*, (2013) for models without explicit microbial biomass.
Models able to reproduce priming effect generally need an explicit microbial biomass
controlling the decomposition (Blagodatsky et al., 2010; Perveen et al., 2014). The priming
scheme used here allows some simplifications in the model structure since an explicit
representation of the microbial biomass is not needed. Furthermore, at equilibrium state (i.e.
when the input rate is constant) the decomposition rate of a first order equation (Eq. (6)) takes
*PE* implicitly into account. When FOC inputs are modified, due to the tree growth for instance,
the *PE* intensity is modified and this effect cannot be represented by classical first order
kinetics. To estimate the importance of priming on SOC storage in the agroforestry plot, the
simulations using first order equations (Eq. (6)) can therefore not be directly compared to the
simulations using the FOC-dependant decomposition rate (Eq. (7)). To estimate the change of
SOC decomposition rate due to priming when trees are planted, the decomposition rate
predicted by Eq. (7) $\left(-k_{HSOC,z} \times (1 - e^{-PE \times FOC_{t,z,d}})\right)$ in the agroforestry plot has to be
compared to the control plot decomposition rate. Thus, to calculate the importance of priming
on SOC storage when trees are planted, we used the decomposition rates calculated following
Eq. (7) in the control plot $\left(-k_{HSOC,z} \times (1 - e^{-PE \times FOC_{t,z,d}})\right)$ and we applied this decomposition
rate to the agroforestry plot as a classical first order kinetics (without the FOC from the control
plot). This simulation corresponded to the absence of priming due to trees in the agroforestry





plot (i.e. decomposition not controlled by the FOC of the agroforestry plot). By difference with
the simulation performed with the full two pools model (Eq. (7)), i.e., taking account of FOC
input and priming, we calculated the priming intensity.

**3 Results**
**3.1 Experimental results**
**3.1.1 Carbon stock in the walnut tree biomass**
The measured aboveground (trunk + branches) and stump carbon stock of 18-year-old walnut
trees are presented in Table 3.

**Table 3.** Carbon stocks in the aboveground biomass and in the stump of 18-year-old walnut

527        trees (110 trees ha$^{-1}$).


| | Tree biomass carbon stock | |
|---|---|---|
| | (kg C tree$^{-1}$) | (t C ha$^{-1}$) |
| Trunk | 55.06 ± 4.35 | 6.06 ± 0.48 |
| Branches | 40.98 ± 7.65 | 4.51 ± 0.84 |
| Stump | 21.21 ± 1.07 | 2.33 ± 0.12 |
| Total | 117.25 ± 8.87 | 12.9 ± 0.98 |

Errors represent standard errors.

**3.1.2 Tree growth**
Tree growth measurements enabled us to fit the following equation that was used in the model:
$$DBH_t \begin{cases} 0.01, & t \leq 3 \\ 0.0157 \times t - 0.0391 \ (R^2 = 0.997) & 3 < t \leq 20 \end{cases} \quad (22)$$

where $DBH_t$ is the diameter at breast height (m) and $t$ represents the time since tree planting
(years).

**3.1.3 Crop yield**



The average annual crop yield in the control plot was $Y_C = 3.79 \pm 0.40$ t DM ha$^{-1}$ for the 14
studied years. In the agroforestry plot, the average relative yield decreased linearly with time
(increasing *DBH*) and was described using the following linear equation (Fig. 2):
$\quad Rel\ Y_{AF_t} = -93.33 \times DBH_t + 100 \qquad (R^2 = 0.12, \qquad p-value = 0.02) \qquad (23)$
where $Rel\ Y_{AF_t}$ is the average relative crop yield (%) in the agroforestry plot compared to the
control plot at year *t*, and $DBH_t$ is the diameter at breast height (m) at year *t*.

In the agroforestry plot, a linear relationship was used to describe the relative crop yield increase
from the tree to the middle of the alley (Fig. 2):
$\quad\quad Rel\ Y_{AF_d} = 4.39 \times d + 64.57 \quad (R^2 = 0.24), \qquad 1 < d \le 6.5 \qquad (24)$
where $Rel\ Y_{AF_d}$ is the relative crop yield (%) in the agroforestry plot at a distance *d* (m) from
the tree compared to the control plot.

Finally, the crop yield in the agroforestry plot was modeled as follows:
$\quad\quad Y_{AF_{t,d}} = Rel\ Y_{AF_t} \times Y_C \times Rel\ Y_{AF_d} \quad (R^2 = 0.19), \qquad 1 < d \le 6.5 \qquad (25)$
where $Y_{AF_{t,d}}$ is the crop yield (t DM ha$^{-1}$) in the agroforestry plot at the year *t* and at a distance
*d* (m) from the tree. Because three linear equations were used to describe the crop yield in the
agroforestry plot, errors were accumulated and we finally came up with a standard
underestimation of the crop yield in the agroforestry plot that we corrected by multiplying our
equation by 1.2.




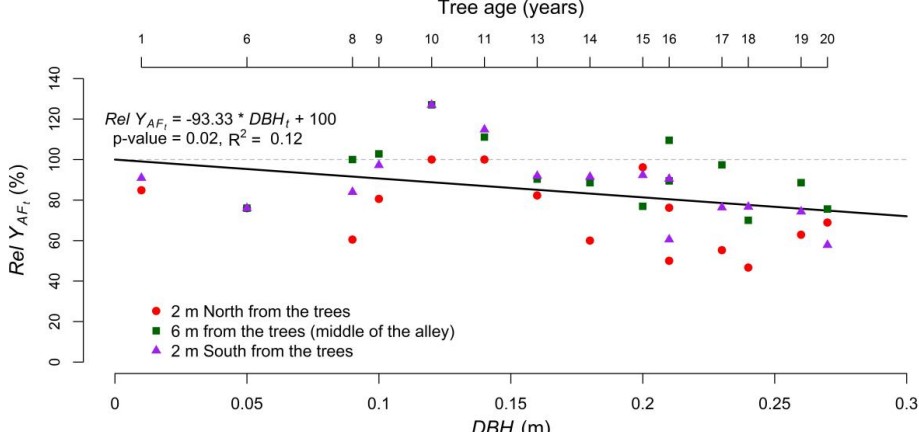

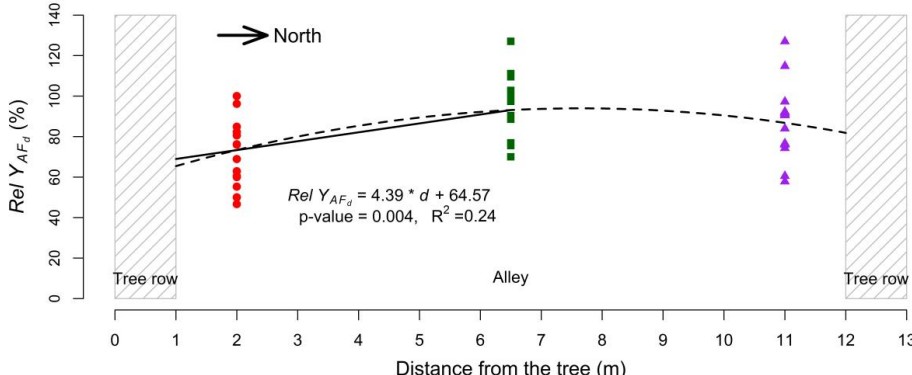


**Fig. 2.** Top: Relative yield ($Rel\ Y_{AF_t}$) in the agroforestry plot compared to the control plot as a

function of tree growth, represented by the diameter at breast height ($DBH$) at year $t$.

Bottom: Relative yield ($Y_{AF_{t,d}}$) as a function of the distance from the tree.


**3.2 Carbon inputs to the FOC pool**

**3.2.1 Leaf litterfall** Total leaf biomass was $8.96 \pm 1.45$ kg DM tree$^{-1}$ and the carbon

concentration of walnut leaves was $449.4 \pm 3.7$ mg C g$^{-1}$ (Table 2). With a density of 110 trees

ha$^{-1}$, leaf litterfall was estimated at $0.73 \pm 0.06$ t C ha$^{-1}$ in 2012 and at the plot scale. The ratio





between leaf biomass and $DBH$ was $0.0277 \pm 0.0024$ t C tree$^{-1}$ m$^{-1}$ or 3.05 t C ha$^{-1}$ m$^{-1}$. The
following linear relationship was therefore used in the model to describe leaf litter C input:
$$L_t = 3.05 \times DBH_t \qquad (26)$$
where $L_t$ is the leaf litter input (t C ha$^{-1}$) at the year $t$, and $DBH_t$ the diameter at breast height
(m) the year $t$.

**3.2.2 Tree fine root C input from mortality**
In 2012, the measured tree fine root biomass was higher in the tree row than in the alley (Table
4). From 0 to 1 m distance from the tree (in the tree row), the tree fine root biomass was
homogeneous and was 1.01 t C ha$^{-1}$ down 2 m depth.

**Table 4.** Walnut tree fine root biomass (t C ha$^{-1}$) as a function of depth and distance from the

582        trees (m).

| | Tree fine root biomass (t C ha$^{-1}$) | | | |
|---|---|---|---|---|
| | Tree row | | Alley | |
| Soil depth (m) | [0, 1] m | ]1, 2.5] m | ]2.5, 4.0] m | ]4.0, 5.5] m |
| 0.0-0.1 | $0.08 \pm 0.01$ | $0.08 \pm 0.01$ | $0.01 \pm 0.00$ | $0.00 \pm 0.00$ |
| 0.1-0.3 | $0.14 \pm 0.02$ | $0.24 \pm 0.02$ | $0.15 \pm 0.02$ | $0.05 \pm 0.01$ |
| 0.3-0.5 | $0.22 \pm 0.02$ | $0.16 \pm 0.02$ | $0.08 \pm 0.01$ | $0.05 \pm 0.01$ |
| 0.5-1.0 | $0.35 \pm 0.04$ | $0.14 \pm 0.01$ | $0.14 \pm 0.01$ | $0.08 \pm 0.01$ |
| 1.0-1.5 | $0.15 \pm 0.02$ | $0.10 \pm 0.01$ | $0.08 \pm 0.01$ | $0.08 \pm 0.01$ |
| 1.5-2.0 | $0.07 \pm 0.01$ | $0.13 \pm 0.01$ | $0.09 \pm 0.01$ | $0.07 \pm 0.01$ |
| Total | $1.01 \pm 0.06$ | $0.84 \pm 0.04$ | $0.55 \pm 0.03$ | $0.34 \pm 0.02$ |

Data modified from Cardinael *et al.*, (2015b). Errors represent standard errors.

In 2012 and in the alley, the tree fine root biomass decreased with increasing distance from the
tree and was represented by an exponential function:
$$TFRB = \begin{cases} 1.01, & 0 \le d \le 1 \\ 1.29 \times e^{-0.28 \times d} \quad (R^2 = 0.90), & 1 < d \le 6.5 \end{cases} \qquad (27)$$



where *TFRB* represents tree fine root biomass down 2 m depth (t C ha$^{-1}$), and $d$ the distance
from the tree (m).

The following linear relationship was used to simulate *TFRB* as a function of tree growth:
$$TFRB_{t,d} = \begin{cases} 3.69 \times DBH_t, & 0 \leq d \leq 1 \\ 4.70 \times DBH_t \times e^{-0.28 \times d}, & 1 < d \leq 6.5 \end{cases} \quad (28)$$
where *TFRB$_t$* represents the tree fine root biomass to 2 m depth (t C ha$^{-1}$) at the year $t$, *DBH$_t$* the
diameter at breast height (m) at the year $t$, and $d$ the distance to the tree (m).

A decreasing exponential function best represented the changing distribution of tree fine roots
within the soil profile with increasing distance to the tree:
$$p_{TFRB,z,d} = \begin{cases} 13.92 \times e^{-1.39 \times z} & (R^2 = 0.68), & 0 \leq d \leq 1 \\ a \times e^{-b \times z}, & & 1 < d \leq 6.5 \end{cases} \quad (29)$$
and
$$a = 10.31 - 1.15 \times d \quad (R^2 = 0.69) \quad (30)$$
$$b = -1.10 + 0.19 \times d \quad (R^2 = 0.51) \quad (31)$$
Finally,
$$p_{TFRB,z,d} = \begin{cases} 13.92 \times e^{-1.39 \times z}, & 0 \leq d \leq 1 \\ (10.31 - 1.15 \times d) \times e^{-(-1.10+0.19 \times d) \times z}, & 1 < d \leq 6.5 \end{cases} \quad (32)$$
where $p_{TFRB,z,d}$ is the proportion (%) of the total tree fine root biomass (*TFRB*) at a given depth
$z$ (m), and at a distance $d$ from the tree (m).

The tree fine root turnover ranged from 1.7 to 2.8 yr$^{-1}$ depending on fine root diameter, with an
average turnover of 2.2 yr$^{-1}$ for fine roots $\leq$ 2 mm and to a depth of 2 m (Germon et al., 2016).

**3.2.3 Aboveground carbon input from the crop**
In the agroforestry plot, the carbon input to the soil from the aboveground crop biomass was:



$$ABC_{crop,t,d} = Y_{AF_{t,d}} \times (straw\ biomass\text{:} crop\ yield) \times C_{straw} \times (1 - export) \qquad (33)$$
where $ABC_{crop,t,d}$ is the aboveground carbon input from the crop (t C ha$^{-1}$) at the year $t$ and
distance $d$ from the tree, $Y_{AF_{t,d}}$ is the agroforestry crop yield. The average ratio between the
straw biomass (t DM ha$^{-1}$) and the crop yield (t DM ha$^{-1}$) equaled $1.03 \pm 0.11$ (n=6). The wheat
straw was exported out of the field after the harvest, but it was estimated that 25% of the straw
biomass was left on the soil, thus *export*=0.75. In the control plot, $Y_{AF_{t,d}}$ was replaced by $Y_C$.

**3.2.4 Belowground carbon input from the crop**
In the agroforestry plot, the belowground crop biomass was represented by:
$$BEC_{crop,t,d} = Y_{AF_{t,d}} \times (shoot\text{:} crop\ yield) \times (root\text{:} shoot) \times C_{root} \qquad (34)$$
where $BEC_{crop,t,d}$ is the belowground crop biomass (t C ha$^{-1}$) at the year $t$ and at a distance $d$
from the tree, $Y_{AF_{t,d}}$ is the agroforestry crop yield. The average ratio between the total crop
aboveground biomass (*shoot*) and the crop yield equaled $2.45 \pm 0.15$ (n=6). In 2012, total fine
root biomass was $2.29 \pm 0.32$ t C ha$^{-1}$ in the control (Table 5).

**Table 5.** Wheat fine root biomass in the agricultural control plot in 2012.

| Soil depth (m) | Wheat fine root biomass | |
| --- | --- | --- |
| | (kg C m$^{-3}$) | (t C ha$^{-1}$) |
| 0.0-0.1 | 0.48 ± 0.05 | 0.48 ± 0.05 |
| 0.1-0.3 | 0.34 ± 0.04 | 0.69 ± 0.09 |
| 0.3-0.5 | 0.22 ± 0.04 | 0.44 ± 0.08 |
| 0.5-1.0 | 0.10 ± 0.04 | 0.52 ± 0.20 |
| 1.0-1.5 | 0.03 ± 0.04 | 0.17 ± 0.19 |
| Total | - | 2.29 ± 0.32 |

Errors represent standard errors.

Therefore, the wheat *root:shoot* ratio equaled $0.79 \pm 0.12$ (n=1). The carbon concentration of
wheat root was $C_{root} = 35.14 \pm 1.90$ mg C g$^{-1}$. In the control plot, $Y_{AF_{t,d}}$ was replaced by $Y_C$.



In 2012, no wheat roots were observed below 1.5 m, and root biomass decreased exponentially
with increasing depth (Table 5). The distribution of crop roots within the soil profile was
described as follows:
$$p_{CRBc,z} = \begin{cases} 26.44 \times e^{-2.59 \times z} & (R^2 = 0.99), \quad z \leq 1.5 \\ 0, & z > 1.5 \end{cases} \quad (35)$$

where $p_{CRBc,z}$ is the proportion (%) of total crop root biomass in the control plot at a given
depth $z$ (m).
The crop root turnover was assumed to be 1 yr$^{-1}$, root mortality occurring mainly after crop
harvest.

**3.2.5 Aboveground and belowground carbon inputs from the tree row herbaceous**
**vegetation**
The distance from the trees had no effect on the above and belowground biomass of the
herbaceous vegetation (data not shown), therefore average values are presented. The summer
aboveground biomass was almost three times higher than in winter, whereas the belowground
biomass was two times higher (Table 6). The total aboveground carbon input was $2.13 \pm 0.14$ t
C ha$^{-1}$ yr$^{-1}$ and the total belowground carbon input was $0.74 \pm 0.05$ t C ha$^{-1}$ yr$^{-1}$ to 0.5 m depth.

**Table 6.** Aboveground and belowground biomass of the herbaceous vegetation in the tree rows.

|  | Soil depth (m) | Herbaceous biomass (t C ha$^{-1}$) | |
| --- | --- | --- | --- |
|  |  | Summer | Winter |
| Aboveground | - | $1.57 \pm 0.11$ | $0.56 \pm 0.09$ |
| Belowground | 0.0-0.1 | $0.22 \pm 0.03$ | $0.17 \pm 0.01$ |
|  | 0.1-0.3 | $0.16 \pm 0.02$ | $0.06 \pm 0.01$ |
|  | 0.3-0.5 | $0.09 \pm 0.04$ | $0.04 \pm 0.01$ |
|  | Total | $0.46 \pm 0.04$ | $0.27 \pm 0.02$ |

Errors represent standard errors.



The belowground carbon input from the tree row vegetation ($BEC_{veg,z}$, t C ha$^{-1}$) at a given depth
$z$ (m) was described by the following equation:
$$BEC_{veg,z} = \begin{cases} 0.44 \times e^{-3.12 \times z}, & z \le 1.5 \\ 0, & z > 1.5 \end{cases} \qquad (36)$$


**3.2.6 Organic carbon inputs and SOC stocks: a synthesis from field measurements**
Tree rows in the agroforestry system received two times more organic carbon (OC) inputs
compared to the control plot (Fig. 3), and 65% more than alleys. Globally, the agroforestry plot
had 41% more OC inputs to the soil than the control plot to 2 m depth (3.80 t C ha$^{-1}$ yr$^{-1}$
compared to 2.69 t C ha$^{-1}$ yr$^{-1}$). In the control plot, 85% of OC inputs are wheat root litters. In
the agroforestry plot, root inputs represent 71% of OC inputs in the alleys, and 50% in the tree
rows.

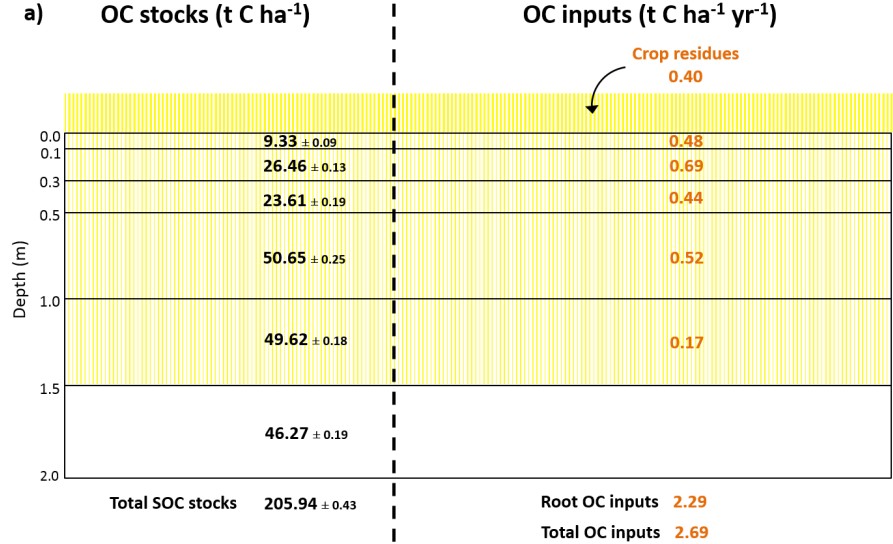





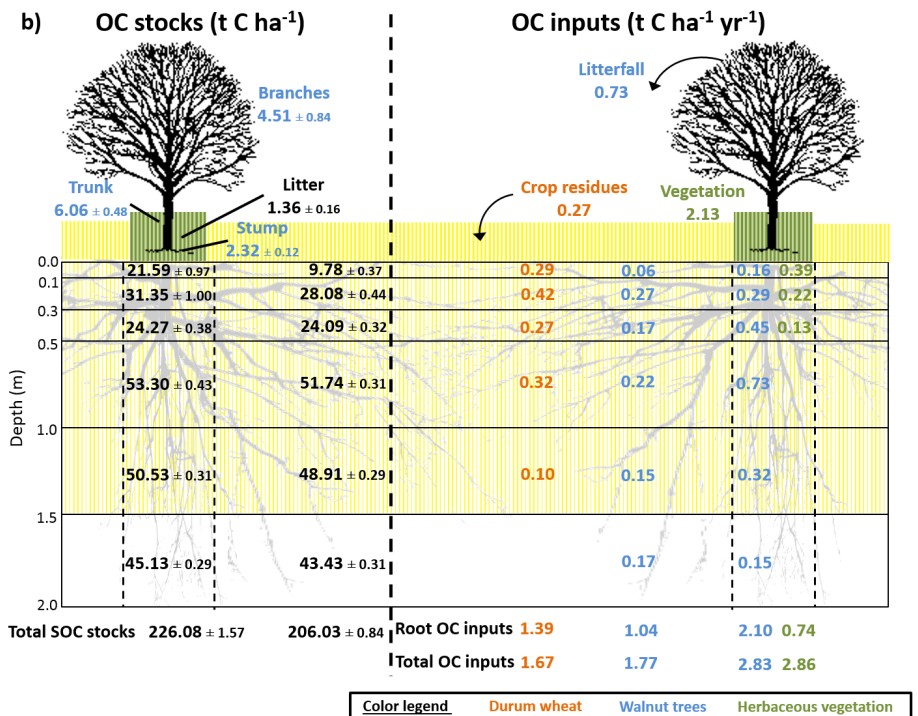


**Fig. 3.** Measured soil organic carbon stocks and organic carbon inputs to the soil a) in the

agricultural control plot, b) in the 18-year-old agroforestry plot. Associated errors are

standard errors. Values are expressed per hectare of land type (control, alley, tree row).

To get the values per hectare of agroforestry, data from alley and tree row have to be

weighted by their respective surface area (i.e., 84% and 16%, respectively) and then

added up. OC: organic carbon; SOC: soil organic carbon. SOC stocks data are issued

from Cardinael *et al.*, (2015a), data of tree root OC inputs are combined from Cardinael

*et al.*, (2015b) and from Germon *et al.*, (2016).


**3.3 HSOC decomposition rate**

The soil incubation experiment showed that the HSOC mineralization rate decreased

exponentially with depth (Fig. S1) and could be described with:



$$k_{HSOC,z} = 6.114 \times e^{-1.37 \times z} \quad (R^2 = 0.76) \qquad (34)$$

where z is the soil depth (m), and where the $a$ (yr$^{-1}$) coefficient ($a = 6.114$) was further optimized
(Table 7).



**Table 7.** Summary of optimized model parameters.

| Model parameter | Meaning | Prior range | Posterior values ± variance (prior values) | | | |
|---|---|---|---|---|---|---|
| | | | 2 pools – without *PE* | 2 pools – with *PE* | 3 pools – without *PE* | |
| $a$ | coefficient from Eq. (8) of the HSOC decomposition (yr$^{-1}$) | 3.65$^{-6}$-3.65 | 0.01e$^{-2}$ ± <10$^{-4}$ (0.01e$^{-2}$) | 0.01e$^{-2}$ ± <10$^{-4}$ (0.01e$^{-2}$) | - | |
| $a_1$ | coefficient from Eq. (8) of the HSOC1 decomposition (yr$^{-1}$) | 3.65$^{-6}$-3.65 | - | - | 0.01e$^{-2}$ ± <10$^{-4}$ (0.01e$^{-2}$) | |
| $a_2$ | coefficient from Eq. (8) of the HSOC2 decomposition (yr$^{-1}$) | 3.65$^{-6}$-3.65 | - | - | 0.83e$^{-2}$ ± 0.17e$^{-2}$ (0.83e$^{-2}$) | |
| $D$ | diffusion coefficient (cm$^2$ yr$^{-1}$) | 1e$^{-6}$-1 | 4.62e$^{-4}$ ± 5.95e$^{-4}$ (9.64e$^{-4}$) | 5.63e$^{-4}$ ± 1.42e$^{-4}$ (9.01e$^{-4}$) | 5.24e$^{-4}$ ± 7.62e$^{-4}$ (9.64e$^{-4}$) | |
| $A$ | advection rate (mm yr$^{-1}$) | 1e$^{-6}$-1 | 21.25e$^{-4}$ ± 5.02e$^{-4}$ (8.54e$^{-4}$) | 6.63e$^{-4}$ ± 2.38e$^{-4}$ (4.27e$^{-4}$) | 21.60e$^{-4}$ ± 2.24e$^{-4}$ (8.54e$^{-4}$) | |
| $h$ | humification yield | 0.01-1 | 0.32 ± <10$^{-4}$ (0.34) | 0.25 ± 1.00e$^{-4}$ (0.13) | 0.34 ± 0.03 (0.34) | |
| $PE$ | priming coefficient | 0.1-160 | - | 9.66 ± 1.49 (102.95) | - | |
| $f_1$ | fraction of decomposed FOC entering the HSOC1 pool | 0-1 | - | - | 0.99 ± 0.18 (0.86) | |
| $f_2$ | fraction of decomposed HSOC1 entering the FOC pool | 0-1 | - | - | 0.94 ± 1.10e$^{-3}$ (0.80) | |





**3.4 Modeling results**
**3.4.1 Optimized parameters and correlation matrix**
The optimized parameters and their prior modes are presented in Table 7. For the two pools
model without priming effect, the most important correlation was observed between $h$ and $A$
which control the humification and the transport by advection. Concerning the two pools model
with priming effect, the most important correlations were observed between $h$ and $PE$ which
controls the effect of the FOC on HSOC decomposition, and between $h$ and $A$. $A$ and $PE$ were
also positively correlated (Fig. S2). For the three pools model, $f_1$ and $f_2$ were by definition
negatively correlated, but $f_2$ and $A$ were also correlated. Considering the method used to
optimize the parameters, these important correlation factors hinder the presentation of the
model output within an envelope. Therefore, we presented the model results using the optimized
parameter without any envelope.

**3.4.2 Modeled SOC stocks**
Observed SOC stocks were not well represented by the two pools model without priming effect,
with RMSE ranging from 1.00 to 1.07 kg C m$^{-3}$ (Fig. 4, Table S1). The model performed better
when the priming effect was taken into account, with RMSE ranging from 0.41 to 0.95 kg C m$^{-}$
$^3$, and the SOC profile was well described. The representation of SOC stocks was not improved
by the inclusion of a third C pool in the model. Globally, the two pools model with priming
effect was the best one, as shown by the BICs (Fig. 4, Table S1). For all models, SOC stocks
below 1 m depth were better described than above SOC stocks (Table S1). The spatial
distribution of SOC storage was also well described (Fig. 5), with a very high SOC stock in the
topsoil layer in the tree row. Most modeled SOC storage in the agroforestry plot was located in
the first 0.2 m depth, and SOC storage was slightly higher in the middle of the alleys than in
the alleys close to the tree rows.

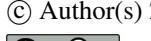







**Fig. 4.** Measured and modeled soil organic carbon contents (kg C m$^{-3}$) in an agricultural control plot and in an 18-year-old silvoarable system with a two pools model without priming effect (no *PE*), with a two pools model with priming effect (*PE*) and with a three pools model without *PE*.






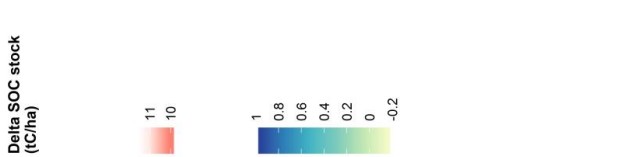

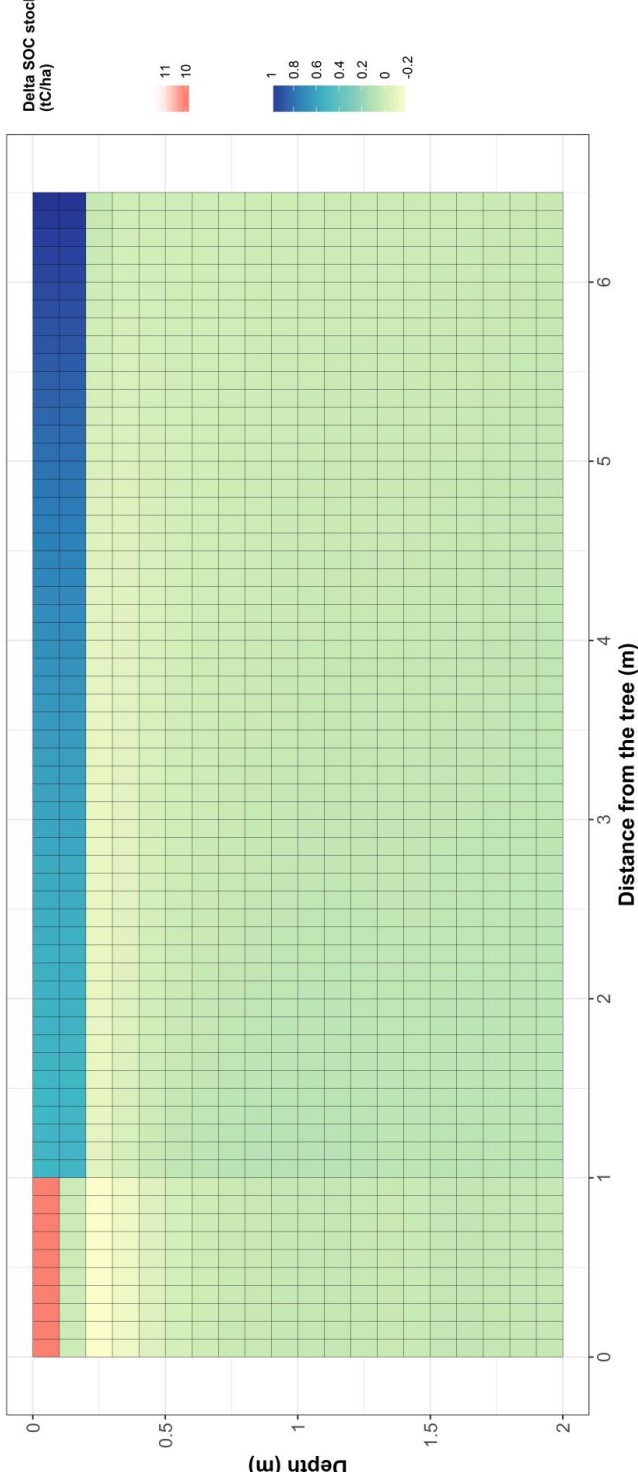


**Fig. 5.** Spatial distribution of additional SOC storage (t C ha$^{-1}$) in an 18-year-old silvoarable system compared to an agricultural control plot and


represented by the two pools model with priming effect.





### 3.4.3 Antagonist effect of priming on SOC storage


The priming effect increases the decomposition rate when more FOC is available. Therefore,
the effect of a C inputs increase on SOC storage in the agroforestry plot might be
counterbalanced by priming. With our model we were able to estimate the contribution of each
driver on SOC storage. The introduction of priming effect in the model reduced the potential
SOC storage due to higher organic inputs in the agroforestry system by 91% in the alley, and
by 76% in the tree rows (Fig. 6). The potential effect of OC inputs alone on SOC storage was
49.12 to 62.77 t C ha$^{-1}$, but the effect of priming on SOC storage was -44.89 to -47.67 t C ha$^{-1}$,
resulting in a modeled SOC storage of 4.23 t C ha$^{-1}$ in the alley and of 15.09 t C ha$^{-1}$ in the tree
row down 2 depth (Fig. 6). The negative effect of priming effect on SOC storage increased with
increasing soil depth (Fig. S3).

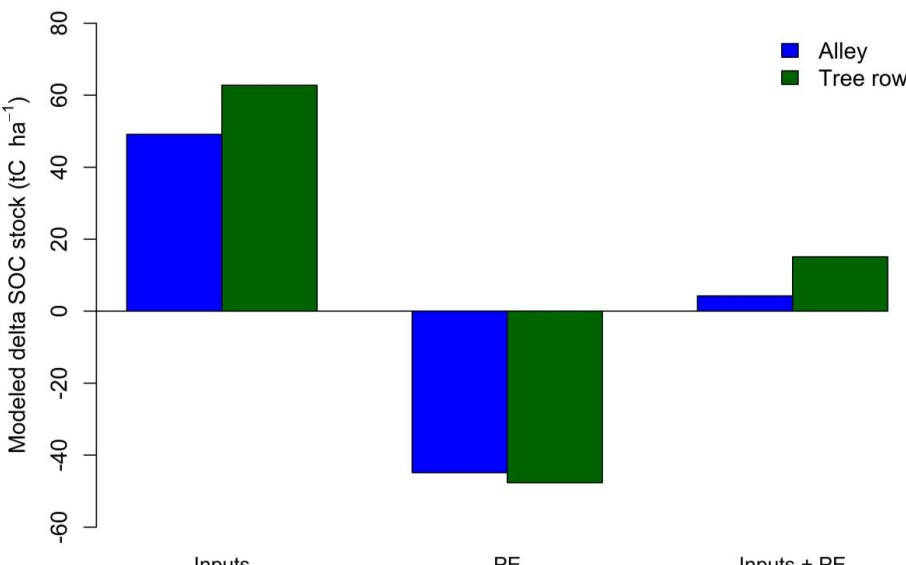


**Fig. 6.** Decoupling the role of C inputs and priming effect (*PE*) on SOC storage in an 18-year-
old silvoarable system down 2 m depth. Inputs: only the input effect is modeled; *PE*:
only the priming effect is modeled; Inputs + *PE*: model prediction with both processes
taken into account.




## 4 Discussion

### 4.1 OC inputs drive SOC storage in agroforestry systems

Increased SOC stocks in the agroforestry plot compared to the control may be explained either
by increased OC inputs, or decreased OC outputs by SOC mineralization, or both. Measured
organic carbon inputs to soil were increased by 40% down to 2m depth in the 18-year-old
agroforestry plot compared to the control plot. Increased OC inputs in agroforestry systems has
been shown in other studies but they were only quantified in the first 20 cm of soil (Oelbermann
et al., 2006; Peichl et al., 2006). This study is therefore the first one also quantifying deep OC
inputs to soil. In this study and due to a lack of data, soil temperature and soil moisture were
considered the same in both plots so that abiotic factors controlling SOC decomposition were
identical. The model was able to well reproduce SOC stocks in the agroforestry plot, suggesting
that OC inputs is the main driver of SOC storage, and that a decrease of SOC mineralisation
due to the agroforestry microclimate is not obvious. Reduced soil temperature is often observed
in agroforestry systems (Clinch et al., 2009; Dubbert et al., 2014), but effect of agroforestry on
soil moisture is much more complex. The soil evaporation is reduced under the trees, but water
is lost through their transpiration (Ilstedt et al., 2016; Ong and Leakey, 1999), and these effects
vary with the distance from the tree (Odhiambo et al., 2001). Moreover, the water infiltration
and the water storage can be increased under the trees after a rainy event (Anderson et al.,
2009). Therefore, the effect of agroforestry on soil moisture is variable in time and space, and
should be investigated more in details. Interactions between soil temperature and soil moisture
on the SOC decomposition are known to be complex (Conant et al., 2011; Moyano et al., 2013;
Sierra et al., 2015) and up to now it is not possible to predict the effect of agroforestry
microclimate on the SOC decomposition rate. A sensitivity analysis performed on these two
boundary conditions showed that the model was not very sensitive to soil temperature and soil

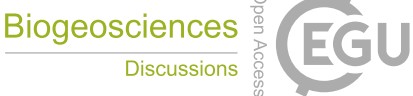



moisture (Fig. S4) suggesting that the potential changes in soil microclimate in the agroforestry
plot are not major drivers of the SOC storage. Furthermore, the SOC decomposition rate could
also be modified due to an absence of soil tillage in the tree rows (Balesdent et al., 1990) or to
an increased aggregate stability (Udawatta et al., 2008) in the topsoil.

**4.2 Representation of SOC spatial heterogeneity in agroforestry systems**
The lateral spatial heterogeneity of SOC stocks in the agroforestry plot was well described by
the model, with higher SOC stocks in the tree rows' topsoil than in the alleys. Inputs from the
herbaceous vegetation had an important impact on SOC storage in this agroforestry system.
The model treated the carbon from this litter as an input to the upper layer of the mineral soil,
in the same way as inputs by roots. Introduction of nitrogen in the model could be further tested
in order to take into account a lower carbon use efficiency due to a lack of nutrients for
microbial growth in this litter. For all models, SOC stocks were better described in the tree rows
than in the alleys. In the alleys, the spatial distribution of organic inputs is more complex and
thus more difficult to model. The tree root system is influenced by the soil tillage and by the
competition with the crop roots, and thus the highest tree fine root density is not observed in
the topsoil but in the 0.3-0.5 m soil layer (Cardinael et al., 2015a). In the model, we were not
able to represent this specific tree root pattern with commonly used mathematical functions,
and tree root profiles were modeled, by default, using a decreasing exponential. Indeed,
piecewise linear functions introduce threshold effects not desirable for transport mechanisms,
especially diffusion. This simplification could partly explain the model overestimation of SOC
stocks in the 0.0-0.1 m layer of the alleys compared to observed data. This result suggests that
it could be useful to couple the CARBOSAF model with a model describing root architecture
and root growth (Dunbabin et al., 2013; Dupuy et al., 2010), using for instance voxel automata
(Mulia et al., 2010). Moreover, the model described a slight increase of SOC stocks in the



middle of the alleys than close to the trees in the alleys. This could be explained by the linear
equation used to describe the crop yield as a function of the distance from the trees, leading to
an overestimation of the crop yield reduction close to the trees. It could also be explained by
the formalism used to model leaf litter distribution in the plot. We considered a homogeneous
distribution of leaf inputs in the agroforestry plot, which was the case in the last years, but
probably not in the first years of the tree growth where leaves might be more concentrated close
to the trees (Thevathasan and Gordon, 1997).
The model also represented a slight SOC storage in the agroforestry plot below 1.0 m depth,
but it was not observed in the field. This could be linked to an overestimation of C input from
tree fine root mortality. Indeed, a constant root turnover was considered along the soil profile,
but several authors reported a decrease of the root turnover with increasing soil depth (Germon
et al., 2016; Hendrick and Pregitzer, 1996; Joslin et al., 2006). However, the sensitivity analysis
showed that the model was not sensitive to this parameter (Fig. S4).

**4.3 Vertical representation of SOC profiles in models**
The best model to represent SOC profiles considered the priming effect. This process can act
in two different ways on the shape of SOC profiles. It has a direct effect on the SOC
mineralization and it therefore modulates the amount of SOC in each soil layer, creating
different SOC gradients. This indirectly affects the mechanisms of C transport within the soil
profile, as shown by a modification of transport coefficients in the case of priming effect (Table
7). Contrary to what was shown by Cardinael *et al.*, (2015c) in long term bare fallows receiving
contrasted organic amendments, the addition of another SOC pool could not surpass the
inclusion of priming effect in terms of model performance. Together with Wutzler &
Reichstein, (2013) and Guenet *et al.*, (2016), this study therefore suggests that implementing



priming effect into SOC models would improve model performances especially when
modelling deep SOC profiles.
We considered here the same transport coefficients for the FOC and HSOC pools, but the
quality and the size of OC particles are different, potentially leading to various movements in
the soil by water fluxes or fauna activity (Lavelle, 1997). Moreover, we considered identical
transport parameters in the agroforestry and in the control plot, but the presence of trees could
modify soil structure, soil water fluxes (Anderson et al., 2009), and the fauna activity (Price
and Gordon, 1999). However, the model was little sensitive to these parameters (Fig. S4).
Further study could investigate the role of different transport coefficients on the description of
SOC profiles.

**4.4 Higher OC inputs or a different quality of OC?**
The introduction of trees in an agricultural field not only modifies the amount of litter residues,
but also their quality. Tree leaves, tree roots, and the herbaceous vegetation from the tree row
have different C:N ratios, lignin and cellulose contents than the crop residues. Recent studies
showed that plant diversity had a positive impact on SOC storage (Lange et al., 2015; Steinbeiss
et al., 2008). One of the hypothesis proposed by the authors is that diverse plant communities
result in more active, more abundant and more diverse microbial communities, increasing
microbial products that can potentially be stabilized. In our model, litter quality is not related
to different SOC pools, but is implicitly taken into account in the FOC decomposition rate,
which is weighted by the respective contribution from the different types of OC inputs. To test
this, we performed a model run considering that all OC inputs in the agroforestry plot were crop
inputs (all FOC decomposition rates equaled wheat decomposition rate), but results were not
significantly different from the one presented here. We then consider that changes in litter
quality in the agroforestry plot did not significantly influence SOC decomposition rates.




### 4.5 Possible limitation of SOC storage by priming effect

Our modelling results showed that the priming effect could considerably reduce the capacity of

soils to store organic carbon. Our study showed that the increase of SOC stocks was not

proportional to OC inputs, especially at depth. This result has often been observed in Free Air

$CO_2$ Enrichment (FACE) experiments. In these experiments, productivity is usually increased

due to $CO_2$ fertilization, but several authors also reported an increase in SOC decomposition

but not linearly linked to the productivity increase (van Groenigen et al., 2014; Sulman et al.,

2014). In this study, the estimation of the priming effect intensity was possible because most

OC inputs to the soil were accurately measured. The modelled intensity of priming effect was

very strong, offsetting 75 to 90% of potential SOC storage due to OC inputs. In a long-term

FACE experiment, Carney *et al.*, (2007) also found that SOC decreased due to priming effect,

offsetting 52% of additional carbon accumulated in aboveground and coarse root biomass. The

priming effect intensity also relies on nutrient availability (Zhang et al., 2013). In agroforestry

systems, tree roots can intercept leached nitrate below the crop rooting zone (Andrianarisoa et

al., 2016), reducing nutrient availability. This beneficial ecosystem service could indirectly

increase the priming effect intensity in deep soil layers.

However, this strong intensity could also partially be linked to the formalism used to simulate

priming effect. This formalism assumes that there is no mineralisation of the SOC in the

absence of fresh OC inputs (no basal respiration). This is a strong hypothesis, but this situation

never occurs since the FOC pool is never empty (data not shown). In the alleys and below the

maximum rooting depth of crops, there are no direct inputs of FOC, but OC is transported in

these deep layers due to transport mechanisms. However, further studies could study the impact

of the priming effect formalism on the estimation of its intensity by using explicit microbial

biomass for instance (Blagodatsky et al., 2010; Perveen et al., 2014).





Finally, root exudates were not quantified in this study. Several authors showed that they could
induce strong priming effects (Bengtson et al., 2012; Keiluweit et al., 2015), but root exudates
are also a source of labile carbon, potentially contributing to stable SOC (Cotrufo et al., 2013).
These opposing effects of root exudates on SOC should be further investigated, especially
concerning the deep roots in agroforestry systems.

**5 Conclusions**
We proposed the first model that simulates soil organic carbon dynamics in agroforestry
accounting for both the whole soil profile and the lateral spatial heterogeneity in agroforestry
plots. This model described reasonably well the measured SOC stocks after 18 years of
agroforestry and SOC distributions with depth. It showed that the increased inputs of fresh
biomass to soil in the agroforestry system explained the observed additional SOC storage and
suggested priming effect as a process controlling SOC stocks in the presence of trees. This
study points out at processes that may be modified by deep rooting trees and deserve further
studies given their potential effects on SOC dynamics, such as additional inputs of C as roots
exudates, or altered soil structure leading to modified SOC transport rates.

**6 Data availability**
The data and the model are freely available upon request and can be obtained by contacting the
author (remi.cardinael@cirad.fr).

**Information about the Supplement**
The Supplement includes the different model performances (Table S1), the potential SOC
decomposition rate as a function of soil depth (Fig. S1), the correlation matrix of optimized



888 parameters (Fig. 8), the decoupling of OC inputs and priming effect as a function of soil depth

889 (Fig. S3), and a sensitivity analysis of the model (Fig. S4).


891 *Acknowledgments.*

892 This study was financed by the French Environment and Energy Management Agency

893 (ADEME), following a call for proposals as part of the REACCTIF program (Research on

894 Climate Change Mitigation in Agriculture and Forestry). This work was part of the funded

895 project AGRIPSOL (Agroforestry for Soil Protection, 1260C0042), coordinated by Agroof.

896 Rémi Cardinael was supported both by ADEME and by La Fondation de France. We thank the

897 farmer, Mr Breton, who allowed us to sample in his field. We are very grateful to our colleagues

898 for their work in the field since the tree planting, especially Jean-François Bourdoncle, Myriam

899 Dauzat, Lydie Dufour, Jonathan Mineau, Alain Sellier and Benoit Suard. We thank colleagues

900 and students who helped us for measurements in the field or in the laboratory, especially Daniel

901 Billiou, Cyril Girardin, Patricia Mahafaka, Agnès Martin, Valérie Pouteau, Alexandre Rosa,

902 and Manon Villeneuve. Finally, we would like to thank Jérôme Balesdent and Pierre Barré for

903 their valuable comments on the modeling part of this work.

904

905 **References**

906 Ahrens, B., Braakhekke, M. C., Guggenberger, G., Schrumpf, M. and Reichstein, M.:

907 Contribution of sorption, DOC transport and microbial interactions to the $^{14}$C age of a soil

908 organic carbon profile: Insights from a calibrated process model, Soil Biol. Biochem., 88, 390–

909 402, 2015.

910 Albrecht, A. and Kandji, S. T.: Carbon sequestration in tropical agroforestry systems, Agric.

911 Ecosyst. Environ., 99, 15–27, 2003.

912 Anderson, S. H., Udawatta, R. P., Seobi, T. and Garrett, H. E.: Soil water content and infiltration



in agroforestry buffer strips, Agrofor. Syst., 75(1), 5–16, 2009.
Andrianarisoa, K., Dufour, L., Bienaime, S., Zeller, B. and Dupraz, C.: The introduction of
hybrid walnut trees (*Juglans nigra* x *regia* cv. NG23) into cropland reduces soil mineral N
content in autumn in southern France, Agrofor. Syst., 90(2), 193–205, 2016.
Baisden, W. T. and Parfitt, R. L.: Bomb $^{14}$C enrichment indicates decadal C pool in deep soil?,
Biogeochemistry, 85, 59–68, , 2007.
Baisden, W. T., Amundson, R., Brenner, D. L., Cook, A. C., Kendall, C. and Harden, J. W.: A
multiisotope C and N modeling analysis of soil organic matter turnover and transport as a
function of soil depth in a California annual grassland soil chronosequence, Global
Biogeochem. Cycles, 16(4), 82-1-82–26, 2002.
Balandier, P. and Dupraz, C.: Growth of widely spaced trees. A case study from young
agroforestry plantations in France, Agrofor. Syst., 43, 151–167, 1999.
Balesdent, J., Mariotti, A. and Boisgontier, D.: Effect of tillage on soil organic carbon
mineralization estimated from $^{13}$C abundance in maize fields, J. Soil Sci., 41(4), 587–596, 1990.
Bambrick, A. D., Whalen, J. K., Bradley, R. L., Cogliastro, A., Gordon, A. M., Olivier, A. and
Thevathasan, N. V: Spatial heterogeneity of soil organic carbon in tree-based intercropping
systems in Quebec and Ontario, Canada, Agrofor. Syst., 79, 343–353, 2010.
Bengtson, P., Barker, J. and Grayston, S. J.: Evidence of a strong coupling between root
exudation, C and N availability, and stimulated SOM decomposition caused by rhizosphere
priming effects, Ecol. Evol., 2(8), 1843–1852, 2012.
Blagodatsky, S., Blagodatskaya, E., Yuyukina, T. and Kuzyakov, Y.: Model of apparent and
real priming effects: Linking microbial activity with soil organic matter decomposition, Soil
Biol. Biochem., 42(8), 1275–1283, 2010.
Braakhekke, M. C., Beer, C., Hoosbeek, M. R., Reichstein, M., Kruijt, B., Schrumpf, M. and





Kabat, P.: SOMPROF: A vertically explicit soil organic matter model, Ecol. Modell., 222(10),
1712–1730, 2011.
Bruun, S., Christensen, B. T., Thomsen, I. K., Jensen, E. S. and Jensen, L. S.: Modeling vertical
movement of organic matter in a soil incubated for 41 years with $^{14}$C labeled straw, Soil Biol.
Biochem., 39(1), 368–371, 2007.
Burgess, P. J., Incoll, L. D., Corry, D. T., Beaton, A. and Hart, B. J.: Poplar (*Populus* spp)
growth and crop yields in a silvoarable experiment at three lowland sites in England, Agrofor.
Syst., 63, 157–169, 2004.
Cardinael, R., Mao, Z., Prieto, I., Stokes, A., Dupraz, C., Kim, J. H. and Jourdan, C.:
Competition with winter crops induces deeper rooting of walnut trees in a Mediterranean alley
cropping agroforestry system, Plant Soil, 391, 219–235, 2015a.
Cardinael, R., Chevallier, T., Barthès, B. G., Saby, N. P. A., Parent, T., Dupraz, C., Bernoux,
M. and Chenu, C.: Impact of alley cropping agroforestry on stocks, forms and spatial
distribution of soil organic carbon - A case study in a Mediterranean context, Geoderma, 259–
260, 288–299, 2015b.
Cardinael, R., Eglin, T., Guenet, B., Neill, C., Houot, S. and Chenu, C.: Is priming effect a
significant process for long-term SOC dynamics? Analysis of a 52-years old experiment,
Biogeochemistry, 123, 203–219, 2015c.
Cardinael, R., Chevallier, T., Cambou, A., Béral, C., Barthès, B. G., Dupraz, C., Durand, C.,
Kouakoua, E. and Chenu, C.: Increased soil organic carbon stocks under agroforestry: A survey
of six different sites in France, Agric. Ecosyst. Environ., 236, 243–255, 2017.
Carney, K. M., Hungate, B. A., Drake, B. G. and Megonigal, J. P.: Altered soil microbial
community at elevated CO2 leads to loss of soil carbon, PNAS, 104(12), 4990–4995, 2007.
Charbonnier, F., le Maire, G., Dreyer, E., Casanoves, F., Christina, M., Dauzat, J., Eitel, J. U.



H., Vaast, P., Vierling, L. A. and Roupsard, O.: Competition for light in heterogeneous
canopies: Application of MAESTRA to a coffee (*Coffea arabica* L.) agroforestry system,
Agric. For. Meteorol., 181, 152–169, 2013.
Chaudhry, A. K., Khan, G. S., Siddiqui, M. T., Akhtar, M. and Aslam, Z.: Effect of arable crops
on the growth of poplar (*Populus deltoides*) tree in agroforestry system, Pakistan J. Agric. Sci.,
40, 82–85, 2003.
Chifflot, V., Bertoni, G., Cabanettes, A. and Gavaland, A.: Beneficial effects of intercropping
on the growth and nitrogen status of young wild cherry and hybrid walnut trees, Agrofor. Syst.,
66(1), 13–21, 2006.
Clinch, R. L., Thevathasan, N. V., Gordon, A. M., Volk, T. A. and Sidders, D.: Biophysical
interactions in a short rotation willow intercropping system in southern Ontario, Canada, Agric.
Ecosyst. Environ., 131(1–2), 61–69, 2009.
Conant, R. T., Ryan, M. G., Ågren, G. I., Birge, H. E., Davidson, E. A., Eliasson, P. E., Evans,
S. E., Frey, S. D., Giardina, C. P., Hopkins, F. M., Hyvönen, R., Kirschbaum, M. U. F.,
Lavallee, J. M., Leifeld, J., Parton, W. J., Megan Steinweg, J., Wallenstein, M. D., Martin
Wetterstedt, J. Å. and Bradford, M. A.: Temperature and soil organic matter decomposition
rates - synthesis of current knowledge and a way forward, Glob. Chang. Biol., 17(11), 3392–

978    3404, 2011.

Cotrufo, M. F., Wallenstein, M. D., Boot, C. M., Denef, K. and Paul, E.: The Microbial
Efficiency-Matrix Stabilization (MEMS) framework integrates plant litter decomposition with
soil organic matter stabilization: do labile plant inputs form stable soil organic matter?, Glob.
Chang. Biol., 19(4), 988–95, 2013.
Davidson, E. A. and Janssens, I. A.: Temperature sensitivity of soil carbon decomposition and
feedbacks to climate change, Nature, 440, 165–173, 2006.





Dimassi, B., Cohan, J.-P., Labreuche, J. and Mary, B.: Changes in soil carbon and nitrogen
following tillage conversion in a long-term experiment in Northern France, Agric. Ecosyst.
Environ., 169, 12–20, 2013.
Dubbert, M., Mosena, A., Piayda, A., Cuntz, M., Correia, A. C., Pereira, J. S. and Werner, C.:
Influence of tree cover on herbaceous layer development and carbon and water fluxes in a
Portuguese cork-oak woodland, Acta Oecologica, 59, 35–45, 2014.
Dufour, L., Metay, A., Talbot, G. and Dupraz, C.: Assessing Light Competition for Cereal
Production in Temperate Agroforestry Systems using Experimentation and Crop Modelling, J.
Agron. Crop Sci., 199(3), 217–227, 2013.
Dunbabin, V. M., Postma, J. A., Schnepf, A., Pagès, L., Javaux, M., Wu, L., Leitner, D., Chen,
Y. L., Rengel, Z. and Diggle, A. J.: Modelling root-soil interactions using three-dimensional
models of root growth, architecture and function, Plant Soil, 372(1–2), 93–124, 2013.
Dupuy, L., Gregory, P. J. and Bengough, A. G.: Root growth models: Towards a new generation
of continuous approaches, J. Exp. Bot., 61(8), 2131–2143, 2010.
Duursma, R.A. and Medlyn, B.E.: MAESPA: a model to study interactions between water
limitation, environmental drivers and vegetation function at tree and stand levels, with an
example application to [$CO_2$] × drought interactions, Geosci. Model Dev., 5, 919–940, 2012.
Eilers, K. G., Debenport, S., Anderson, S. and Fierer, N.: Digging deeper to find unique
microbial communities: The strong effect of depth on the structure of bacterial and archaeal
communities in soil, Soil Biol. Biochem., 50, 58–65, 2012.
Eissenstat, D. M. and Yanai, R. D.: The Ecology of Root Lifespan, Adv. Ecol. Res., 27, 1–60,

1006  1997.

Ellert, B. H. and Bettany, J. R.: Calculation of organic matter and nutrients stored in soils under
contrasting management regimes, Can. J. Soil Sci., 75, 529–538, 1995.

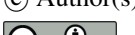



Elzein, A. and Balesdent, J.: Mechanistic simulation of vertical distribution of carbon
concentrations and residence times in soils, Soil Sci. Soc. Am. J., 59, 1328–1335, 1995.
Fierer, N., Schimel, J. P. and Holden, P. A.: Variations in microbial community composition
through two soil depth profiles, Soil Biol. Biochem., 35(1), 167–176, 2003.
Fontaine, S., Barot, S., Barré, P., Bdioui, N., Mary, B. and Rumpel, C.: Stability of organic
carbon in deep soil layers controlled by fresh carbon supply, Nature, 450, 277–281, 2007.
Germon, A., Cardinael, R., Prieto, I., Mao, Z., Kim, J. H., Stokes, A., Dupraz, C., Laclau, J.-P.
and Jourdan, C.: Unexpected phenology and lifespan of shallow and deep fine roots of walnut
trees grown in a silvoarable Mediterranean agroforestry system, Plant Soil, 401, 409–426, 2016.
Graves, A. R., Burgess, P. J., Palma, J. H. N., Herzog, F., Moreno, G., Bertomeu, M., Dupraz,
C., Liagre, F., Keesman, K., van der Werf, W., de Nooy, A. K. and van den Briel, J. P.:
Development and application of bio-economic modelling to compare silvoarable, arable, and
forestry systems in three European countries, Ecol. Eng., 29(4), 434–449, 2007.
Graves, A. R., Burgess, P. J., Palma, J., Keesman, K. J., van der Werf, W., Dupraz, C., van
Keulen, H., Herzog, F. and Mayus, M.: Implementation and calibration of the parameter-sparse
Yield-SAFE model to predict production and land equivalent ratio in mixed tree and crop
systems under two contrasting production situations in Europe, Ecol. Modell., 221, 1744–1756,

1026    2010.

van Groenigen, K. J., Qi, X., Osenberg, C. W., Luo, Y. and Hungate, B. A.: Faster
decomposition under increased atmospheric $CO_2$ limits soil carbon storage, Science,
344(6183), 508–9, 2014.
Guenet, B., Eglin, T., Vasilyeva, N., Peylin, P., Ciais, P. and Chenu, C.: The relative importance
of decomposition and transport mechanisms in accounting for soil organic carbon profiles,
Biogeosciences, 10(4), 2379–2392, 2013.



Guenet, B., Moyano, F. E., Peylin, P., Ciais, P. and Janssens, I. A.: Towards a representation
of priming on soil carbon decomposition in the global land biosphere model ORCHIDEE
(version 1.9.5.2), Geosci. Model Dev., 9, 841–855, 2016.
Haile, S. G., Nair, V. D. and Nair, P. K. R.: Contribution of trees to carbon storage in soils of
silvopastoral systems in Florida, USA, Glob. Chang. Biol., 16, 427–438, 2010.
Hendrick, R. L. and Pregitzer, K. S.: Temporal and depth-related patterns of fine root dynamics
in northern hardwood forests, J. Ecol., 84, 167–176, 1996.
Howlett, D. S., Moreno, G., Mosquera Losada, M. R., Nair, P. K. R. and Nair, V. D.: Soil
carbon storage as influenced by tree cover in the Dehesa cork oak silvopasture of central-
western Spain, J. Environ. Monit., 13(7), 1897–904, 2011.
Ilstedt, U., Bargués Tobella, A., Bazié, H. R., Bayala, J., Verbeeten, E., Nyberg, G., Sanou, J.,
Benegas, L., Murdiyarso, D., Laudon, H., Sheil, D. and Malmer, A.: Intermediate tree cover
can maximize groundwater recharge in the seasonally dry tropics, Sci. Rep., 6, 21930, 2016.
IUSS Working Group WRB: World Reference Base for Soil Resources 2006, first update 2007.
World Soil Resources Reports No. 103. FAO, Rome., 2007.
Jobbagy, E. G. and Jackson, R. B.: The vertical distribution of soil organic carbon and its
relation to climate and vegetation, Ecol. Appl., 10, 423–436, 2000.
Joslin, J. D., Gaudinski, J. B., Torn, M. S., Riley, W. J. and Hanson, P. J.: Fine-root turnover
patterns and their relationship to root diameter and soil depth in a [14]C-labeled hardwood forest,
New Phytol., 172, 523–535, 2006.
Kätterer, T., Bolinder, M. A., Andrén, O., Kirchmann, H. and Menichetti, L.: Roots contribute
more to refractory soil organic matter than above-ground crop residues, as revealed by a long-
term field experiment, Agric. Ecosyst. Environ., 141, 184–192, 2011.
Keiluweit, M., Bougoure, J. J., Nico, P. S., Pett-Ridge, J., Weber, P. K. and Kleber, M.: Mineral



protection of soil carbon counteracted by root exudates, Nat. Clim. Chang., 5, 588-595, 2015.
Kim, D.-G., Kirschbaum, M. U. F. and Beedy, T. L.: Carbon sequestration and net emissions
of CH4 and N2O under agroforestry: Synthesizing available data and suggestions for future
studies, Agric. Ecosyst. Environ., 226, 65–78, 2016.
Koarashi, J., Hockaday, W. C., Masiello, C. A. and Trumbore, S. E.: Dynamics of decadally
cycling carbon in subsurface soils, J. Geophys. Res., 117, 1–13, 2012.
Koven, C. D., Riley, W. J., Subin, Z. M., Tang, J. Y., Torn, M. S., Collins, W. D., Bonan, G.
B., Lawrence, D. M. and Swenson, S. C.: The effect of vertically resolved soil biogeochemistry
and alternate soil C and N models on C dynamics of CLM4, Biogeosciences, 10(11), 7109–

1066    7131, 2013.

Lange, M., Eisenhauer, N., Sierra, C. A., Bessler, H., Engels, C., Griffiths, R. I., Mellado-
Vázquez, P. G., Malik, A. A., Roy, J., Scheu, S., Steinbeiss, S., Thomson, B. C., Trumbore, S.
E. and Gleixner, G.: Plant diversity increases soil microbial activity and soil carbon storage,
Nat. Commun., 6, 6707, 2015.
Lavelle, P.: Faunal activities and soil processes: adaptative strategy that determine ecosystem
function., 1997.
Li, F., Meng, P., Fu, D. and Wang, B.: Light distribution, photosynthetic rate and yield in a
Paulownia-wheat intercropping system in China, Agrofor. Syst., 74(2), 163–172, 2008.
Lorenz, K. and Lal, R.: Soil organic carbon sequestration in agroforestry systems. A review,
Agron. Sustain. Dev., 34, 443–454, 2014.
Luedeling, E., Smethurst, P. J., Baudron, F., Bayala, J., Huth, N. I., van Noordwijk, M., Ong,
C. K., Mulia, R., Lusiana, B., Muthuri, C. and Sinclair, F. L.: Field-scale modeling of tree-crop
interactions: Challenges and development needs, Agric. Syst., 142, 51–69, 2016.
Mead, R. and Willey, R. W.: The concept of a "land equivalent ratio" and advantages in yields



from intercropping, Exp. Agric., 16(3), 217–228, 1980.
Moreno, G., Obrador, J. J., Cubera, E. and Dupraz, C.: Fine root distribution in Dehesas of
central-western Spain, Plant Soil, 277(1–2), 153–162, 2005.
Moyano, F. E., Vasilyeva, N., Bouckaert, L., Cook, F., Craine, J., Curiel Yuste, J., Don, A.,
Epron, D., Formanek, P., Franzluebbers, A., Ilstedt, U., Kätterer, T., Orchard, V., Reichstein,
M., Rey, A., Ruamps, L., Subke, J. A., Thomsen, I. K. and Chenu, C.: The moisture response
of soil heterotrophic respiration: Interaction with soil properties, Biogeosciences, 9, 1173–

1088    1182, 2012.

Moyano, F. E., Manzoni, S. and Chenu, C.: Responses of soil heterotrophic respiration to
moisture availability: An exploration of processes and models, Soil Biol. Biochem., 59, 72–85,

1091    2013.

Mulia, R. and Dupraz, C.: Unusual fine root distributions of two deciduous tree species in
southern France: What consequences for modelling of tree root dynamics?, Plant Soil, 281, 71–

1094    85, 2006.

Mulia, R., Dupraz, C. and van Noordwijk, M.: Reconciling root plasticity and architectural
ground rules in tree root growth models with voxel automata, Plant Soil, 337(1–2), 77–92, 2010.
Nair, P. K.: An introduction to agroforestry, Kluwer, Dordrecht, The Netherlands., 1993.
Nair, P. K. R.: Classification of agroforestry systems, Agrofor. Syst., 3(2), 97–128, 1985.
van Noordwijk, M. and Lusiana, B.: WaNuLCAS, a model of water, nutrient and light capture
in agroforestry systems, Agrofor. Syst., 43, 217–242, 1999.
Odhiambo, H. O., Ong, C. K., Deans, J. D., Wilson, J., Khan, A. A. H. and Sprent, J. I.: Roots,
soil water and crop yield: Tree crop interactions in a semi-arid agroforestry system in Kenya,
Plant Soil, 235(2), 221–233, 2001.
Oelbermann, M. and Voroney, R. P.: And evaluation of the century model to predict soil



organic carbon: examples from Costa Rica and Canada, Agrofor. Syst., 82, 37–50, 2011.
Oelbermann, M., Voroney, R. P. and Gordon, A. M.: Carbon sequestration in tropical and
temperate agroforestry systems: a review with examples from Costa Rica and southern Canada,
Agric. Ecosyst. Environ., 104, 359–377, 2004.
Oelbermann, M., Voroney, R. P., Thevathasan, N. V., Gordon, A. M., Kass, D. C. L. and
Schlönvoigt, A. M.: Soil carbon dynamics and residue stabilization in a Costa Rican and
southern Canadian alley cropping system, Agrofor. Syst., 68(1), 27–36, 2006.
Ong, C. K. and Leakey, R. R. B.: Why tree-crop interactions in agroforestry appear at odds with
tree-grass interactions in tropical savannahs, Agrofor. Syst., 45(1–3), 109–129, 1999.
Parton, W. J., Schimel, D. S., Cole, C. V and Ojima, D. S.: Analysis of factors controlling soil
organic matter levels in great plains grasslands, Soil Sci. Soc. Am. J., 51, 1173–1179, 1987.
Peichl, M., Thevathasan, N. V, Gordon, A. M., Huss, J. and Abohassan, R. A.: Carbon
sequestration potentials in temperate tree-based intercropping systems, southern Ontario,
Canada, Agrofor. Syst., 66, 243–257, 2006.
Perveen, N., Barot, S., Alvarez, G., Klumpp, K., Martin, R., Rapaport, A., Herfurth, D.,
Louault, F. and Fontaine, S.: Priming effect and microbial diversity in ecosystem functioning
and response to global change: A modeling approach using the SYMPHONY model, Glob.
Chang. Biol., 20(4), 1174–1190, 2014.
Price, G. W. and Gordon, A. M.: Spatial and temporal distribution of earthworms in a temperate
intercropping system in southern Ontario, Canada, Agrofor. Syst., 44, 141–149, 1999.
Prieto, I., Roumet, C., Cardinael, R., Kim, J., Maeght, J.-L., Mao, Z., Portillo, N.,
Thammahacksa, C., Dupraz, C., Jourdan, C., Pierret, A., Roupsard, O. and Stokes, A.: Root
functional parameters along a land-use gradient: evidence of a community-level economics
spectrum, J. Ecol., 103, 361–373, 2015.



Prieto, I., Stokes, A. and Roumet, C.: Root functional parameters predict fine root
decomposability at the community level, J. Ecol., 104, 725–733, 2016.
R Development Core Team: R: A language and environment for statistical computing, 2013.
Rasse, D. P., Mulder, J., Moni, C. and Chenu, C.: Carbon turnover kinetics with depth in a
French loamy soil, Soil Sci. Soc. Am. J., 70, 2097–2105, 2006.
Salomé, C., Nunan, N., Pouteau, V., Lerch, T. Z. and Chenu, C.: Carbon dynamics in topsoil
and in subsoil may be controlled by different regulatory mechanisms, Glob. Chang. Biol., 16,
416–426, 2010.
Santaren, D., Peylin, P., Viovy, N. and Ciais, P.: Optimizing a process-based ecosystem model
with eddy-covariance flux measurements: A pine forest in southern France, Global
Biogeochem. Cycles, 21, 1–15, 2007.
Shahzad, T., Chenu, C., Genet, P., Barot, S., Perveen, N., Mougin, C. and Fontaine, S.:
Contribution of exudates, arbuscular mycorrhizal fungi and litter depositions to the rhizosphere
priming effect induced by grassland species, Soil Biol. Biochem., 80, 146–155, 2015.
Sierra, C. A., Trumbore, S. E., Davidson, E. A., Vicca, S. and Janssens, I.: Sensitivity of
decomposition rates of soil organic matter with respect to simultaneous changes in temperature
and moisture, J. Adv. Model. Earth Syst., 7, 335–356, 2015.
Somarriba, E.: Revisiting the past: an essay on agroforestry definition, Agrofor. Syst., 19(3),
233–240, 1992.
Steinbeiss, S., Beßler, H., Engels, C., Temperton, V. M., Buchmann, N., Roscher, C.,
Kreutziger, Y., Baade, J., Habekost, M. and Gleixner, G.: Plant diversity positively affects
short-term soil carbon storage in experimental grasslands, Glob. Chang. Biol., 14(12), 2937–

1151    2949, 2008.

Sulman, B. N., Phillips, R. P., Oishi, A. C., Shevliakova, E. and Pacala, S. W.: Microbe-driven



turnover offsets mineral-mediated storage of soil carbon under elevated CO2, Nat. Clim.
Chang., 4, 1099–1102, 2014.
Taghizadeh-Toosi, A., Christensen, B. T., Hutchings, N. J., Vejlin, J., Kätterer, T., Glendining,
M. and Olesen, J. E.: C-TOOL: A simple model for simulating whole-profile carbon storage in
temperate agricultural soils, Ecol. Modell., 292, 11–25, 2014.
Talbot, G.: L'intégration spatiale et temporelle du partage des ressources dans un système
agroforestier noyers-céréales: une clef pour en comprendre la productivité ?, PhD Dissertation,
Université Montpellier II., 2011.
Tarantola, A.: Inverse problem theory: methods for data fitting and model parameter estimation,
edited by Elsevier., 1987.
Tarantola, A.: Inverse Problem Theory and Methods for Model Parameter Estimation, edited
by SIAM., 2005.
Thevathasan, N. V. and Gordon, A. M.: Poplar leaf biomass distribution and nitrogen dynamics
in a poplar-barley intercropped system in southern Ontario, Canada, Agrofor. Syst., 37(1995),
79–90, 1997.
Udawatta, R. P., Kremer, R. J., Adamson, B. W. and Anderson, S. H.: Variations in soil
aggregate stability and enzyme activities in a temperate agroforestry practice, Appl. Soil Ecol.,
39(2), 153–160, 2008.
Virto, I., Barré, P., Burlot, A. and Chenu, C.: Carbon input differences as the main factor
explaining the variability in soil organic C storage in no-tilled compared to inversion tilled
agrosystems, Biogeochemistry, 108, 17–26, 2012.
van der Werf, W., Keesman, K., Burgess, P., Graves, A., Pilbeam, D., Incoll, L. D., Metselaar,
K., Mayus, M., Stappers, R., van Keulen, H., Palma, J. and Dupraz, C.: Yield-SAFE: A
parameter-sparse, process-based dynamic model for predicting resource capture, growth, and



production in agroforestry systems, Ecol. Eng., 29(4), 419–433, 2007.
Wutzler, T. and Reichstein, M.: Colimitation of decomposition by substrate and decomposers
- a comparison of model formulations, Biogeosciences, 5, 749–759, 2008.
Wutzler, T. and Reichstein, M.: Priming and substrate quality interactions in soil organic matter
models, Biogeosciences, 10(3), 2089–2103, 2013.
Yin, R. and He, Q.: The spatial and temporal effects of paulownia intercropping: The case of
northern China, Agrofor. Syst., 37, 91–109, 1997.
Zhang, W., Wang, X. and Wang, S.: Addition of external organic carbon and native soil organic
carbon decomposition: a meta-analysis., PLoS One, 8(2), e54779, 2013.













