# Peer review of "High organic inputs explain shallow and deep SOC storage in a long-term agroforestry system – Combining experimental and modeling approaches."

_Biogeosciences, 2017_

## Referee Comment (RC1) · T. Wutzler (Referee) · 19 Apr 2017

The study presents a new model development and calibration to an interesting horizontally heterogeneous system. It is based on an impressive compiled data set of observations and derivations of relevant inputs and state variables to compare. The main conclusion is that the observed increases in SOM stocks in an agroforest system are due to higher litter input compared to an agricultural control. The modelling exercise is interesting to the soil modelling community, and to the community researching interactions of vegetation components and management. I state several main points

followed by more detailed comments.

**1 Main points**

1) One calibration aspect of the study that convinces me (and probably other readers as well) of the validity of the study is currently not well highlighted. The model was calibrated to the control plot only. Despite of simplifying assumptions on similarities in climate and vertical transport between the control and the agroforesty system, the model predicted the differing C-stocks in tree rows and alleys and its depth distribution well. This is a strong validation.

2) The quantification of the priming effect (PE) seems to be a bit complicated with running the no-PE model variant with a decomposition rate that was calibrated with the PE-model variant. To my opinion there are more straightforward quantifications already in the data (see detailed comments). I suggest highlighting the result that the priming model variant in Fig 4 was able to capture the depth distribution of C-stocks while no-PE model variant did not.

3) While the mathematical model is well described, information is missing on the solution of the forward model, i.e. the solution of the presented partial-differential equation given a set of parameters. Which method has been used? What was the spatial grid, the same grid as the measurements? Was this grid sufficient to represent the steep concentration gradient in the top soil? Have different grid sizes been tested?

4) To my understanding of the study, the increased C stocks at the walnut tree lines are explained in a big part to increase of the above-ground carbon input by the herbaceous summer vegetation between trees (Fig. 3). I would like to read some discussion on this point. Was there an organic layer?

[Figure]

**2   Detailed comments**

L 412: Instead of interpolating parameters of several fits, I suggest fitting a single equation to the entire dataset with an additional variable "distance to tree" and parameters a and b depend on this distance. However, the simplified procedure here seems to work and this point does not affect the conclusions.

L 444: Please specify exactly which observations and which predictions have been used for calibration.

Table 7: The prior knowledge in eq. 19 was specified as normal distribution. Table 7 instead reports a range of values instead of a mean and a variance ($x_b$ and diagonal of $P_b$ in equation 1). Moreover many ranges span several orders of magnitudes suggesting that the parameters should be log-transformed before estimation. Where does the variance of the posterior come from? And what is the meaning of "prior values" in the posterior column?

Table 7: Where did the prior information come from? Are these uninformative priors or does it affect the results if you take different priors?

Eq 21: Please explain the derivation. Usually the $BIC = ln(n)k - 2log(L)$, which involves the Likelihood instead of the mean squared deviation. From a Bayesian perspective $-2log(L) \propto J_{data}(p)$ , where $J_{data}$ is the first term of J of eq. 19 (excluding the prior term).

L 478: Please, clarify terminology of spin-up vs model calibration. To my understanding you calibrated 4 or 5 parameters depending on the three model variants so that equilibrium stocks, i.e. simulations after 5000 years, were close to observed C-stocks (n=?) of the control plot in 2013. I suggest putting this content to the calibration section.

L 508: This derivation of the effect of priming is hard to grasp. To my opinion its more straightforward is compare predictions of the PE-variant model versus the non-

PE variant; each consistently calibrated and applied for prediction:

- Effects of litter inputs: predictions of no-priming variant only: agroforestry stocks vs control stocks

- Combined effect: prediction of the priming model variant only: at agroforestry plot versus the control plot

- Effects of priming only: prediction of the priming model variant versus the predictions of the no-priming variant for the agroforestry system

Since the profile was not matched well with the no-priming model one can focus on sums.

Fig 3: Please, note that the largest above ground input comes from herbaceous vegetation. Is this an important aspect for C-stocks of the agroforestry system?

L698 (3.4.2): Please, remind the reader that C-stocks of the agroforestry plot were not part of model calibration (that used the control plot only) but are used here for validation.

Fig. 4: This is a nice demonstration of priming formulation being able to match the depth-shape. Although uncertainty of the mean (standard error) is low due to the high sample number, you may add the standard deviation across 93 measurements in order to get an impression of the variability.

I would like to see a figure, where C-depth profiles can be compared between cases without being dispersed across facets. Maybe zoom in to 5 to 15 stock range.

Fig. 5: Please, use a color scale with a clear zero.

Fig. 6 Please, add difference in measured stocks to the "Inputs+PE" column for comparison.

L 753: Suggest: "Despite of these simplifying assumptions, the model calibrated to the control plot was able to ..."

---

## Referee Comment (RC2) · Anonymous Referee #2 · 28 May 2017

This is a comprehensive study that uses an impressive set of field data to build a model for exploring agroforestry impacts on soil organic carbon (SOC). The topic is of interest and fits the scope of the journal. The combination of both field and modeling data is a key strength of this paper and provides interesting results regarding the spatial distribution of SOC in an agroforestry system. The modeling further highlights the potential negative impacts of priming on SOC storage. The methodology, results and most of the interpretation is sound. I therefore recommend this manuscript may be published after addressing the concerns and comments outlined below.

Major comments: 1) Due to lack of data, the authors assume that 'soil temperature and soil moisture conditions were the same in the agroforestry tree rows, alleys and in the control plot (L388ff)'. Given the otherwise extensive data collection at this site it is surprising that these key variables have not been measured. As the authors acknowledge at various places, the impact of agroforestry on the SOC is primarily a result of the altered soil abiotic conditions. In my view the lack of these data hamper the understanding of the true controls and mechanisms responsible for change in SOC in the agroforestry system compared to the agricultural control field.

The sensitivity analysis performed by the authors in an attempt to address this limitation cannot replace the missing information on soil abiotic controls since it merely reflects the model sensitivity to these parameters rather than their actual control on SOC.

This shortcoming also limits some of the discussion. In my view, the related conclusions that 'that OC inputs is the main driver of SOC storage (L752)', that 'a decrease of SOC mineralization due to the agroforestry microclimate is not obvious (L753)' and that 'soil microclimate in the agroforestry plot are not major drivers of the SOC storage (L766)' are therefore not justified.

2) The SOC stock is the product of C concentration per unit soil multiplied by the amount of soil per volume (i.e. bulk density). The study however is entirely focused on explaining changes in SOC due to changes in C concentration (as a result of C input/output) whereas changes in bulk density are not reported. It therefore remains unclear what the separate roles of changes in C concentration and bulk density are in controlling the changes in the total SOC stock (L743ff). While the authors acknowledge that the presence of trees (roots) could modify soil structure (L820), the effects of tree planting on such physical soil properties and subsequently SOC stocks are not well addressed in this study.

3) The authors argue that the two pools model with priming effect was the best one, as shown by the BICs (Fig. 4, Table S1) (L704). However this is not true for the

agroforestry alley which had a similar BIC and RMSE than the noPE model in Fig.4. Since the alley covers most of the area in an agroforesty system, this indicates that the priming effect might be overall less significant for this system as proposed by the authors.

4) Overall I find that the ms is too long, especially the method section is exhaustive (16 pages incl. Figures and Tables) but also parts of the results could be condensed. Given that the compilation of the C stock data is not a primary study goal (L118ff), I suggest that methods and results related to these data could be considerably shortened and partly moved into the supplementary part or refer to by references. For instance, data shown in Table 4 is already published (Cardinael et al., (2015b)) and thus there is no need show this Table once more. Section 3.1 and 3.2, specifically the equations developed here should be moved to the Method or Supplementary section. Details of Section 2.7 could also be moved to the Supplementary part.

Minor comments: Line 658: Here and at other places the authors use the word 'globally' which seems inappropriate in the given context.

L706: 'The spatial distribution of SOC storage was also well described (Fig. 5)' – I disagree, Fig.5 shows the 'additional' SOC in the agroforestry system relative to control but not the absolute amount of SOC storage.

L725: 'The priming effect increases the decomposition rate when more FOC is available' – provide a reference for this statement or use past tense to indicate that this is a result from this study.

L772, 797, 873: At the several places the authors refer to 'the model' while several models (or model variations) were used in this study. Please clarify in each case which of the models (model variation) is meant when referring to one specific model.

Figure 4: It would be helpful to add separate legends to the middle and right column sub-figures in Fig 4; also how is it possible that the model PE follows the measured
SOC profile most closely but results in similar BIC than the noPE model?

---

## Author Comment (AC1) · 25 Jun 2017

Referee 1

The study presents a new model development and calibration to an interesting horizontally heterogeneous system. It is based on an impressive compiled data set of observations and derivations of relevant inputs and state variables to compare. The main conclusion is that the observed increases in SOM stocks in an agroforest system are due to higher litter input compared to an agricultural control. The modelling exercise is interesting to the soil modelling community, and to the community researching interactions of vegetation components and management. I state several main points followed by more detailed comments.

Response: We thank you for your interest in our work, we really appreciated your comments and suggestions. We tried to take into account all your comments and corrected the manuscript on the requested points.

1 Main points

1) One calibration aspect of the study that convinces me (and probably other readers as well) of the validity of the study is currently not well highlighted. The model was calibrated to the control plot only. Despite of simplifying assumptions on similarities in climate and vertical transport between the control and the agroforesty system, the model predicted the differing C-stocks in tree rows and alleys and its depth distribution well. This is a strong validation.

Response: Thanks for this rewarding comment. We better highlighted this result in the abstract "The model was calibrated to the control plot only. . .The model was strongly validated, describing properly the measured SOC stocks and distribution with depth in agroforestry tree rows and alleys". (P2L30-34), but also in the discussion part "Despite these simplifying assumptions on similarities in climate but also on vertical transport between the control and the agroforestry system, the model calibrated to the control plot was able to reproduce SOC stocks in tree rows and alleys and its depth distribution well. This strong validation also suggests that OC inputs is the main driver of SOC storage, and that a potential effect of agroforestry microclimate on SOC mineralization is of minor importance" (P40L732-737).

2) The quantification of the priming effect (PE) seems to be a bit complicated with running the no-PE model variant with a decomposition rate that was calibrated with the PE-model variant. To my opinion there are more straightforward quantifications already in the data (see detailed comments). I suggest highlighting the result that the

priming model variant in Fig 4 was able to capture the depth distribution of C-stocks while no-PE model variant did not.

Response: We tried to better describe how the PE intensity was quantified (P25-27L548-594) (see below) and why we chose this calculation method. We also highlighted in the abstract the fact that only the PE model was able to describe SOC profiles "Moreover, only a priming effect variant of the model was able to capture the depth distribution of SOC stocks" (P2L36-37).

3) While the mathematical model is well described, information is missing on the solution of the forward model, i.e. the solution of the presented partial-differential equation given a set of parameters. Which method has been used? What was the spatial grid, the same grid as the measurements? Was this grid sufficient to represent the steep concentration gradient in the top soil? Have different grid sizes been tested?

Response: Partial-differential equations were solved using the R package deSolve and the ode.1D method (Soetaert et al., 2010) (P27L596-597). The spatial grid was as close as possible to the measurements. Due to some field difficulties, the sampling grid is not totally regular but the modelling grid is. We indeed implicitly assumed this resolution to be sufficient to represent the steep concentration but we did not deeply evaluated the effect of different grid size but if really needed we can provide an analysis in the supplementary material

4) To my understanding of the study, the increased C stocks at the walnut tree lines are explained in a big part to increase of the above-ground carbon input by the herbaceous summer vegetation between trees (Fig. 3). I would like to read some discussion on this point. Was there an organic layer?

Response: Yes, this is absolutely true, the herbaceous vegetation growing between trees in the tree rows plays an important role on SOC storage. This was very much suggested by previous works on SOC storage in these systems (Cardinael et al., 2015, 2017), but proven here with the quantification of OC inputs. We now discussed this

point more into details: "The increased SOC stocks in the tree rows were explained in a big part by an important above-ground carbon input (2.13 t C ha-1 yr-1) by the herbaceous vegetation between trees. This result had already been suggested by Cardinael et al., (2015b) and by Cardinael et al., (2017) who showed that even young agroforestry systems could store SOC in the tree rows while trees are still very small. These "grass strips" indirectly introduced by the tree planting in parallel tree rows have a major impact on SOC stocks of agroforestry systems" (P41L758-764). As commonly observed on grass strips, there was a very thin organic layer (maximum 0.3 cm thick), but not permanent during the season. Climatic conditions are very favorable for litter decomposition there, and we therefore assumed that this thin organic layer did not significantly change moisture and temperature conditions for the below mineral soil.

2 Detailed comments

L 412: Instead of interpolating parameters of several fits, I suggest fitting a single equation to the entire dataset with an additional variable "distance to tree" and parameters a and b depend on this distance. However, the simplified procedure here seems to work and this point does not affect the conclusions.

Response: Yes, this is indeed another possibility. As we were able to well reproduce root profiles with this simplified method, we think it is not really necessary to look for another equation as it would indeed not change the conclusions.

L 444: Please specify exactly which observations and which predictions have been used for calibration.

Response: We used SOC stocks measured in 2013 in the control plot (observations) and predicted SOC stocks (predictions) for the calibration. These stocks were considered at equilibrium (P25L551-553).

Table 7: The prior knowledge in eq. 19 was specified as normal distribution. Table 7 instead reports a range of values instead of a mean and a variance (xb and diagonal

of Pb in equation 1). Moreover many ranges span several orders of magnitudes suggesting that the parameters should be log-transformed before estimation. Where does the variance of the posterior come from? And what is the meaning of "prior values" in the posterior column? Table 7: Where did the prior information come from? Are these uninformative priors or does it affect the results if you take different priors?

Response: We acknowledge that this point was not clear enough. The optimization procedure that we used is sensitive to local minima. We therefore performed 30 optimization procedures starting with different parameter prior values to check that the results did not correspond to a local minimum. The prior range presented in Table 7 represents the range in which prior values were sampled for the 30 optimizations, it is therefore normal that they span several orders of magnitudes. The prior values presented in brackets in the posterior column represent the prior values that minimized the $J(x)$ value. The variance of the posterior is based on Santaren et al., 2007 (GBC 21, GB2013). The BFGS algorithm does not directly calculate variance of posteriors. To obtain them, we quantified the variance using the curvature cost function at its minimum once it was reached. We clarified it in the text: "To determine an optimal set of parameters which minimizes $J(x)$, we used the BFGS gradient-based algorithm (Tarantola, 1987). For each model variant, we performed 30 optimizations starting with different parameter prior values to check that the results did not correspond to a local minimum. As the BFGS algorithm does not directly calculate the variance of posteriors, they were quantified using the curvature cost function at its minimum once it was reached (Santaren et al., 2007)." (P26L571-576), and in the Table 7 (now Table 5) footnote: "The prior range represents the range in which prior values were sampled for the 30 optimizations per model variant. The prior values presented in brackets in the posterior column represent the prior values that minimized the $J(x)$ value (Eq. (34))." (P32L660-661).

Eq 21: Please explain the derivation. Usually the BIC = $\ln(n)k - 2\log(L)$, which involves the Likelihood instead of the mean squared deviation. From a Bayesian perspective

-2log(L) $\alpha$ Jdata(p) , where Jdata is the first term of J of eq. 19 (excluding the prior term).

Response: Here, we used the MSD to estimate the maximum likelihood. This is indeed not the classical BIC. This approach is similar to Manzoni et al., 2012 (SBB 50, 66-76) who used the residual sum of square to estimate the maximum likelihood. We rephrase to clarify: "where N is the number of observations, MSD is the mean squared deviation used to estimate the maximum likelihood, and k is the number of model parameters" (P26L585-586).

L 478: Please, clarify terminology of spin-up vs model calibration. To my understanding you calibrated 4 or 5 parameters depending on the three model variants so that equilibrium stocks, i.e. simulations after 5000 years, were close to observed C-stocks (n=?) of the control plot in 2013. I suggest putting this content to the calibration section.

Response: We moved this paragraph to the optimization procedure section and we clarified the terminology of spin-up vs model calibration: "These four or five parameters were calibrated so that equilibrium SOC stocks, i.e. after 5000 years of simulation, equaled SOC stocks of the control plot in 2013. The associated uncertainty was estimated with the 93 soil cores sampled in the control plot (see section 2.2.1). Due to a lack of relevant data, we assumed that the climate and the land use were the same for the last 5000 years, and that SOC stocks in the control plot were at equilibrium at the time of measurement. Therefore, SOC stocks at the end of the 5000 years of simulation equaled SOC stocks in the control plot. Three different calibrations were performed, corresponding to the three different models that were used: one calibration with the two pools model without the priming effect, one calibration with the two pools model with the priming effect, and one calibration with the three pools model" (P25L548-557). "SOC pools were initialized after a spin-up of 5000 years in the control plot. At t0, SOC stocks in the agroforestry plot therefore equaled SOC stocks of the control plot" (P27L592-594).

L 508: This derivation of the effect of priming is hard to grasp. To my opinion its more straightforward is compare predictions of the PE-variant model versus the non-PE variant; each consistently calibrated and applied for prediction: • Effects of litter inputs: predictions of no-priming variant only: agroforestry stocks vs control stocks • Combined effect: prediction of the priming model variant only: at agroforestry plot versus the control plot • Effects of priming only: prediction of the priming model variant versus the predictions of the no-priming variant for the agroforestry system Since the profile was not matched well with the no-priming model one can focus on sums.

Response: We agree that the calculation was not straightforward and we clarified it in the new version (see below). Nevertheless, we consider our calculation as the most correct even though it is a bit complex to understand it. Indeed, we can not directly compare the different versions of the model to calculate priming because the decomposition rate of a classical first order kinetics takes implicitly into account a fixed fraction of decomposition due to priming. In all situations, there are regular inputs inducing priming and when we optimized the decomposition rate parameter in the control plot we implicitly represented this priming but at a fixed rate. Therefore comparing the different versions of the model would not estimate the priming in the agroforestry plots. "Furthermore, at equilibrium state (i.e. when the input rate is constant) the decomposition rate of a first order equation (Eq. (6)) takes PE implicitly into account. Indeed, when FOC enters the system, there is an induced priming, a constant FOC input rate therefore induces a constant priming. This means that when we optimized the decomposition rate parameter in the control plot, we implicitly represented this priming but at a fixed rate. When FOC inputs are modified, due to the tree growth for instance, the PE intensity is modified and this effect cannot be represented by classical first order kinetics." (P27L607-612). "To estimate the change of SOC decomposition rate due to priming when trees are planted, the decomposition fluxes predicted by Eq. (7) (ãĂŰ-kãĂŮ_(HSOC,z)×(1-eˆ(-PE×ãĂŰFOCãĂŮ_(t,z,d) ) )) in the agroforestry plot must be compared to the fluxes in agroforestry plot using the decomposition from the control

plot calculated by Eq. (7) with ãĂŰFOCãĂŮ_(t,z,d) corresponding to the FOC inputs in the control plot. Thus, to calculate the importance of priming on SOC storage when trees are planted, we used the decomposition rates calculated following Eq. (7) in the control plot and we applied this decomposition rate to the agroforestry plot as a classical first order kinetics (without the FOC control, i.e. ãĂŰk_new= kãĂŮ_(HSOC,z)×(1-eˆ(-PE×ãĂŰFOCãĂŮ_(t,z,d) ) ) with FOCt,z,d fixed constant)". (P28L616-625).

Fig 3: Please, note that the largest above ground input comes from herbaceous vegetation. Is this an important aspect for C-stocks of the agroforestry system?

Response: Yes, this is definitely an important aspect for C-stocks in the agroforestry system. We added the following sentence to the result section: "In the agroforestry plot, the largest aboveground OC input to the soil comes from the herbaceous vegetation, and not from the trees" (P28-29L636-637).

L698 (3.4.2): Please, remind the reader that C-stocks of the agroforestry plot were not part of model calibration (that used the control plot only) but are used here for validation.

Response: As suggested, we added the following sentence at the beginning of the section: "As a reminder, SOC stocks of the agroforestry plot were not part of model calibration (that used the control plot only) but were used here for validation" (P33L677-678).

Fig. 4: This is a nice demonstration of priming formulation being able to match the depth-shape. Although uncertainty of the mean (standard error) is low due to the high sample number, you may add the standard deviation across 93 measurements in order to get an impression of the variability. I would like to see a figure, where C-depth profiles can be compared between cases without being dispersed across facets. Maybe zoom in to 5 to 15 stock range.

Response: Yes, the uncertainty of the mean is extremely low for measured SOC stocks,

as suggested we instead added the standard deviation of measurements (P36L693-694). Concerning the C-depth profiles of Fig 4., this was actually our first idea. But SOC profiles are extremely close, especially between the control and the alleys, and the figure was very messy. We would therefore prefer to stick to this presentation, which is much clearer, even if we have to compare different facets.

Fig. 5: Please, use a color scale with a clear zero.

Response: We changed the color scale as requested. We also added a 2D graph of modeled control and agroforestry SOC stocks (P37).

Fig. 6 Please, add difference in measured stocks to the "Inputs+PE" column for comparison.

Response: Thanks for this suggestion, it was done (P39).

L 753: Suggest: "Despite of these simplifying assumptions, the model calibrated to the control plot was able to ..."

Response: This sentence was changed as follows: "Despite these simplifying assumptions on similarities in climate but also on vertical transport between the control and the agroforestry system, the model calibrated to the control plot was able to reproduce SOC stocks in tree rows and alleys and its depth distribution well. This strong validation also suggests that OC inputs is the main driver of SOC storage at this site, and that a potential effect of agroforestry microclimate on SOC mineralization is of minor importance" (P40L732-737).

Please also note the supplement to this comment:
http://www.biogeosciences-discuss.net/bg-2017-125/bg-2017-125-AC1-supplement.pdf

---

## Author Comment (AC2) · 25 Jun 2017

Referee 2

This is a comprehensive study that uses an impressive set of field data to build a model for exploring agroforestry impacts on soil organic carbon (SOC). The topic is of interest and fits the scope of the journal. The combination of both field and modeling data is a key strength of this paper and provides interesting results regarding the spatial distribution of SOC in an agroforestry system. The modeling further highlights the

potential negative impacts of priming on SOC storage. The methodology, results and most of the interpretation is sound. I therefore recommend this manuscript may be published after addressing the concerns and comments outlined below.

Response: We thank you for your interest and you positive comments on our work.

Major comments:

1) Due to lack of data, the authors assume that 'soil temperature and soil moisture conditions were the same in the agroforestry tree rows, alleys and in the control plot (L388ff)'. Given the otherwise extensive data collection at this site it is surprising that these key variables have not been measured. As the authors acknowledge at various places, the impact of agroforestry on the SOC is primarily a result of the altered soil abiotic conditions. In my view the lack of these data hamper the understanding of the true controls and mechanisms responsible for change in SOC in the agroforestry system compared to the agricultural control field.

Response: We agree with this comment, it is a pity that soil moisture and soil temperature sensors have not been installed in both fields, and on the long term. But this trial was first established to study crop yield and tree growth in association, and questions on SOC dynamics came very recently. In May 2013 (late Spring, about 15 days after the last rain), we sampled 40 soil cores in the tree rows, 60 in the alleys, and 93 in the control, and we measured soil moisture on 23 of them. Soil cores were first taken in the agroforestry plot, and then in the control plot, under sunny conditions for both plots. The results showed that soil moisture was lower in the first 40 cm of soil in the control plot, but that there was no difference below (see Figure in the Supplement) During the last sampling day in the agroforestry plot, some cores were also taken in the control plot, and the same difference in terms of soil moisture was observed, suggesting that the lowest soil moisture in the control plot were not due to the sampling delay. Trees in the agroforestry plot probably slowed down the soil evaporation due to the shade. Most of the additional SOC storage in the agroforestry plot was observed in the topsoil. The lower topsoil soil moisture observed in the control in May 2013 would induce a reduction of SOC decomposition compared to the agroforestry plot, and then would reduce the observed SOC storage. But we can not conclude with this punctual observation, this phenomenon probably alternates during the season. For instance, we could hypothesize that in summer, deep soil will be drier in the agroforestry plot than in the control due to tree water absorption. Due to these uncertainties, we thought it was wiser to consider a mean annual soil temperature and moisture identical in both fields.

The sensitivity analysis performed by the authors in an attempt to address this limitation cannot replace the missing information on soil abiotic controls since it merely reflects the model sensitivity to these parameters rather than their actual control on SOC. This shortcoming also limits some of the discussion. In my view, the related conclusions that 'that OC inputs is the main driver of SOC storage (L752)', that 'a decrease of SOC mineralization due to the agroforestry microclimate is not obvious (L753)' and that 'soil microclimate in the agroforestry plot are not major drivers of the SOC storage (L766)' are therefore not justified.

Response: We tried to detail but also nuance our conclusions as suggested: "Despite these simplifying assumptions on similarities in microclimate but also on vertical transport between the control and the agroforestry system, the model calibrated to the control plot was able to reproduce SOC stocks in tree rows and alleys and its depth distribution well. This strong validation also suggests that OC inputs is the main driver of SOC storage at this site, and that a potential effect of agroforestry microclimate on SOC mineralization is of minor importance." (P40L732-737). "A sensitivity analysis performed on these two boundary conditions showed that the model was not very sensitive to soil temperature and soil moisture (Fig. S4), but the real effect of these two parameters on SOC dynamics under agroforestry systems should be better investigated in future studies" (P41L747-750).

2) The SOC stock is the product of C concentration per unit soil multiplied by the amount of soil per volume (i.e. bulk density). The study however is entirely focused

on explaining changes in SOC due to changes in C concentration (as a result of C input/output) whereas changes in bulk density are not reported. It therefore remains unclear what the separate roles of changes in C concentration and bulk density are in controlling the changes in the total SOC stock (L743ff). While the authors acknowledge that the presence of trees (roots) could modify soil structure (L820), the effects of tree planting on such physical soil properties and subsequently SOC stocks are not well addressed in this study.

Response: This is a very relevant point, soil bulk densities were only lower in the topsoil in the tree rows compared to the alleys and to the control plot. Bulk densities were published earlier (Cardinael et al., 2015b) and thus not reported here. In the model, we used the measured soil bulk densities for the control, tree rows and alleys from Cardinael et al., (2015b) (P8L172). We then expressed SOC stocks on an equivalent soil mass basis, and not at fixed depth. Therefore, the change in bulk density was implicitly taken into account in this study.

3) The authors argue that the two pools model with priming effect was the best one, as shown by the BICs (Fig. 4, Table S1) (L704). However this is not true for the agroforestry alley which had a similar BIC and RMSE than the noPE model in Fig.4. Since the alley covers most of the area in an agroforesty system, this indicates that the priming effect might be overall less significant for this system as proposed by the authors.

Response: In this case, alleys occupied 84% of the agroforestry area. The BIC and RMSE were lower with the PE model than with the noPE model as indicated in Fig. 4, but we acknowledge the difference is small. In the alleys, the first soil layer (0-10 cm) was worse represented by the PE model than by the noPE model. As the BIC is calculated on the whole profile, this bad fit impacts the BIC even if the PE model performs much better for the other soil layers, this is well shown in Table S2. We therefore think that the PE is need to represent correctly the profile in the alley.

4) Overall I find that the ms is too long, especially the method section is exhaustive (16 pages incl. Figures and Tables) but also parts of the results could be condensed. Given that the compilation of the C stock data is not a primary study goal (L118ff), I suggest that methods and results related to these data could be considerably shortened and partly moved into the supplementary part or refer to by references. For instance, data shown in Table 4 is already published (Cardinael et al., (2015b)) and thus there is no need show this Table once more. Section 3.1 and 3.2, specifically the equations developed here should be moved to the Method or Supplementary section. Details of Section 2.7 could also be moved to the Supplementary part.

Response: We agree that the MS is very long, which is mainly due to the huge amount of data that are compiled here. Moreover, it also includes the differential equations of a new model, which we think are better to be presented in the main manuscript than in the supplementary. We however performed the following changes in order to shorten the description and facilitate comprehension: Tree fine root biomass data previously shown in Table 4 were moved to the supplementary part (Table S1). Moreover, Section 3.1.1 "Carbon stock in the walnut tree biomass" and Table 3 were deleted as results were already presented in Fig. 3. Section 3.1.2 "Tree growth" was moved to the Method part and merged with section 2.6.2 "Interpolation of tree growth" (P18L400-403). Section 3.1.3 "Crop yield" was also moved to the Method part and merged with section 2.6.5 "Aboveground and belowground input from the crop" (P20-24L457-524). Section 3.2.1"Leaf litterfall" was moved to the Method part and merged with section 2.6.3 "Change of tree litterfall over time" (P18L407-415). Section 3.2.2 "Tree fine root C input from mortality" was moved to the Method part and merged with section 2.6.4 "Tree fine root C input from mortality" (P18-20L418-454). Section 3.2.3 "Aboveground carbon input from the crop" and section 3.2.4 were moved to the Method part and merged with section "Aboveground and belowground input from the crop" (P20-24L457-524). Section 3.2.5 "Aboveground and belowground carbon inputs from the tree row herbaceous vegetation" was moved to the Method part and merged with section 2.6.6 "Aboveground and belowground input from herbaceous vegetation in the tree

rows" (P24L528-541). Section 3.2.6 "Organic carbon inputs and SOC stocks: a synthesis from field measurements" was however kept in the Results (now Section 3.1).

Minor comments:

Line 658: Here and at other places the authors use the word 'globally' which seems inappropriate in the given context.

Response: "Globally" was replace by "Overall" (P28L634 and P33L683).

L706: 'The spatial distribution of SOC storage was also well described (Fig. 5)' – I disagree, Fig.5 shows the 'additional' SOC in the agroforestry system relative to control but not the absolute amount of SOC storage.

Response: We now also added to Figure 5 both SOC stocks in the control and in the agroforestry plot. This sentence was modified to "The spatial distribution of SOC stocks and of additional SOC storage was also well described (Fig. 5), with a very high additional SOC stock storage in the topsoil layer in the tree row" (P33L685-687).

L725: 'The priming effect increases the decomposition rate when more FOC is available'– provide a reference for this statement or use past tense to indicate that this is a result from this study.

Response: We added the reference of Fontaine et al., (2007) (P39L706-707).

L772, 797, 873: At the several places the authors refer to 'the model' while several models (or model variations) were used in this study. Please clarify in each case which of the models (model variation) is meant when referring to one specific model.

Response: We now specified it "the two pools model with priming effect" (P41L756, P42L787 and P45L864).

Figure 4: It would be helpful to add separate legends to the middle and right column sub-figures in Fig 4; also how is it possible that the model PE follows the measured SOC profile most closely but results in similar BIC than the noPE model?

Response: As suggested, we added a common legend for all sub-figures at the bottom of Fig 4 (P35). In the alleys, The PE model has almost similar BIC than the noPE model only because the first soil layer (0-10 cm) was worse represented by the PE model: (Model – Measures)^2 = 7.71 compared to 1.28 kg/m3 for the noPE model. As the BIC is calculated on the whole profile, this bad fit impacts the BIC even if the PE model performs much better for the other soil layers, this is well shown in Table S2.

Please also note the supplement to this comment:
http://www.biogeosciences-discuss.net/bg-2017-125/bg-2017-125-AC2-supplement.pdf

---

## Author Response (AR2)

**Dr Rémi Cardinael**
CIRAD – UR AIDA
Avenue d'Agropolis
34398 Montpellier, France
+33 (0)4.67.61.56.88
Mail: remi.cardinael@cirad.fr

Dear Editor,

We acknowledge receipt of the review concerning our manuscript (bg-2017-125) entitled "**High organic inputs explain shallow and deep SOC storage in a long-term agroforestry system – Combining experimental and modeling approaches.**" by Rémi Cardinael, Bertrand Guenet, Tiphaine Chevallier, Christian Dupraz, Thomas Cozzi and Claire Chenu.

One of the reviewer was very satisfied by the corrections we made and recommended this manuscript to be published in its current form. The other reviewer however requested additional modifications.

We took most of its comments into consideration in this version. However, some of them were directly related to the optimization method we used. Our optimization method is different from the method he suggested but it is also recognized as a valid optimization method (Santaren et al., 2007, Barré et al., 2010). If it could be interesting to compare both methods as suggested by the reviewer, we consider that it is a different work. It would require a long time and a complete reorganization of the paper. Optimization procedures would have to be done again, as well as model runs and outputs' representation. Therefore, we decided not to do the requested changes concerning the optimization method. On the contrary, we better explained our method and detailed why it was valid.

Other comments were related to the grid resolution. The purpose of the work was not to provide a fully flexible model, but to have a model adapted to our data and able to answer our questions. We tried a finer grid resolution and it did not change much the results. We agreed that adding details on the resolution would dilute the main message of the paper.

From the beginning of the reviewing process, both reviewers agreed on the originality and on the impressive measured data set from the oldest agroforestry trial in Europe (18 years old): crop yields, tree growth, all fresh organic litter inputs to soil, organic carbon stocks down to 2 m depth to show that the increase of organic inputs to soil was able to explain SOC storage in these systems. These results have global interest since long-term studies on SOC sequestration are scarce, especially in temperate agroforestry systems. We showed that agroforestry systems are part of the solution to help mitigating climate change. These systems are also food producing systems (compared to afforestation for instance), and are therefore of high interest for the "4 per 1000, *Soils for food security and climate* Initiative" (http://4p1000.org/). Modeling SOC dynamics in these spatially heterogeneous systems is necessary and new.

We still think that Biogeosciences is the most appropriate journal for such a work, and we hope we were able to convince the Editor and the reviewers of the quality of this work.

Best regards,

Rémi Cardinael, on behalf of the co-authors

Montpellier, November 14th, 2017

**ASSOCIATE EDITOR**

*I have now received two reviewer reports. While one of the reviewers is satisfied with the revision, serious concerns are expressed by the other that need further consideration. Please make carefully sure to extend and change the manuscript accordingly. Report all changes to the manuscript with clear reference to the reviewer comments they address.*

**Response:** We took into account several points raised by the first referee, but we were not able to fully answer all of them.

**NOTE for the reviewers: Pages and lines refer here to the marked-up manuscript version.**

**REPORT 1, REFEREE 2**

*I find that the revised manuscript is greatly improved. The study provides a valuable contribution to improving our understanding of soil carbon dynamics in agroforestry systems. The authors have adequately addressed all of my previous comments. I therefore recommend that this manuscript may be published in its current form.*

**Response:** Thank you for your comments.

**REPORT 2, REFEREE 1**

*The presentation of the manuscript improved compared to the previous version, but not enough. Some required additional analysis has not been done. With the better representation several issues became more clear that need to be tackled. The manuscript should only be published with more serious work on reacting on the review comments.*

**Response:**

*The study reports changes across the entire soil profile. However, changes seem to be most important in the top 20cm and in the tree alleys (See strange legend in Fig 4 bottom with a change in magnitude for one level). When reading only about the aggregated results, it implied to me different conclusions than reading the full paper in detail. The authors did some small notes in reacting on review requests for these details but did very little adjustments in presenting the overall results. I suggest reporting and discussing results independently in a) tree-alleys, b) top 20cm in rows c) below 20cm d) aggregated space.*

**Response:** The reviewer is correct, most of the additional SOC storage was located in the topsoil, especially in the tree rows. We tried to reorganize the results and the discussion parts as proposed by the reviewer, not only the ones concerning SOC stocks, but also the ones concerning OC inputs:

"In the alleys of the18-year-old agroforestry system, measured organic carbon (OC) inputs from the crop residues and roots were reduced compared to the control plot due a lower crop yield (Fig. 3). This reduction in crop OC inputs was offset by OC inputs from the tree roots and tree litterfall. Total root OC inputs in the alleys (crop + tree roots) and in the control plot (crop roots) were very similar, respectively 2.43 and 2.29 t C ha$^{-1}$ yr$^{-1}$. Alleys received 0.60 t C ha$^{-1}$ yr$^{-1}$ more of total aboveground biomass (crop residues + tree litterfall) than the control, which was added to the plough layer. Tree rows received 2.35 t C ha$^{-1}$ yr$^{-1}$ more C inputs in the first 0.3 m of soil compared to the control plot, mainly from the herbaceous vegetation. Down the whole soil profile, tree rows received two times more OC inputs compared to the control plot (Fig. 3), and 65% more than alleys. Overall, the agroforestry plot had 41% more OC inputs to the soil than the control plot to 2 m depth (3.80 t C ha$^{-1}$ yr$^{-1}$ compared to 2.69 t C ha$^{-1}$ yr$^{-1}$)" (P30-L646-657).

"In the first 0.3 m of soil, SOC stocks were significantly higher in the alleys than in the control plot, but the difference was small (2.1 ± 0.6 t C ha$^{-1}$). Between 0.3 and 1.0 m, the difference of SOC stocks was smaller but still significant. However, between 1 and 2 m depth, SOC stocks were significantly lower in the alleys than in the control. As a consequence, there was no significant difference of total SOC stocks between the two locations down the whole soil profile. In the tree rows, topsoil organic carbon stocks (0.0-0.3 m) were much higher than in the control (+ 17.0 ± 1.4 t C ha$^{-1}$). SOC stocks in the tree rows were also significantly higher than in the control down 1.5 m depth, but the difference was small. The opposite was observed between 1.5 and 2.0 m depth. Delta of total SOC stocks between the tree rows and the control plot was 20 ± 1.6 t C ha$^{-1}$. At the plot scale, total SOC stocks were significantly higher in the agroforestry plot compared to the control plot down 2 m depth (+ 3.3 ± 0.9 t C ha$^{-1}$)" (P31-L661-671).

"In the alleys, higher SOC stocks in the topsoil could be explained by inputs from litterfall and tree roots despite a decrease in crop inputs. Most of additional SOC storage in the agroforestry plot was found in the topsoil in the tree rows. The same distribution was observed for OC inputs to the soil. Inputs from the herbaceous vegetation had an important impact on SOC storage. The increased SOC stocks in the tree rows were largely explained by an important above-ground carbon input (2.13 t C ha-1 yr-1) by the herbaceous vegetation between trees. This result had already been suggested by Cardinael et al., (2015b) and by Cardinael et al., (2017) who showed that even young agroforestry systems could store SOC in the tree rows while trees are still very small. These "grass strips" indirectly introduced by the tree planting in parallel tree rows have a major impact on SOC stocks of agroforestry systems. Increased SOC stocks below the plough layer could be explained by higher root inputs, but these inputs could also have contributed to decrease SOC stocks below 1.5 m due to priming effect. At the plot scale, measured organic carbon inputs to the soil were increased by 40% (+1.1 t C ha$^{-1}$ yr$^{-1}$) down 2 m depth in the 18-year-old agroforestry plot compared to the control plot, resulting in increased SOC stocks of 3.3 t C ha$^{-1}$" (P40-L754-759).

*The study claims performing a Bayesian analysis because it applies prior information on model parameters. However, exact prior information is still not presented (e.g. diagonal values of P_b in eq. 34) let alone motivated sufficiently. Moreover prediction uncertainty is not presented and reasons for not doing that are very weak (see detailed comments). Uncertainties and full probability distributions are essential components of a Bayesian analysis. I again require to*

*present and motivate the priors and to discuss predictive uncertainties. I still recommend using log-transformed parameters (and hence log-normal prior information) where prior parameter ranges span several orders of magnitude.*

**Response:** Here we used a BFGS algorithm for some reasons detailed below. This method does not allow considering uncertainties in the prior and is sensitive to local minimum. To overcome this difficulty, we launched the optimization procedure 30 times with different set of priors and we chose the set of prior/posterior corresponding to the lowest cost function value. This approach has been used previously by Santaren et al., 2007 or Barré et al. 2010. The different set of priors were randomly chosen within the range we fixed (see table 5, page 34). Following the comment of the referee, we now presented the prior information on a log base. However, we did not perform the optimization on a log base as the $f_1$ and $f_2$ parameters were also optimized and comprised between 0 and 1.

*The presentation of the numerical solution is now sufficient to be understood. There is still works to do on notation (see detailed comments). Why did the authors not check that the resolution of the spatial grid is sufficient? Why should the spatial modeling grid be regular? In the used Soetart's ode.1d code there is no such constraint. With looking at the results, I again strongly recommend repeating the analysis with a finer vertical resolution at the top 30cm.*

**Response:** We used a regular grid because most of the information we had on the boundaries' conditions were at a regular vertical resolution of 10 cm or 20 cm (root biomass, clay, bulk density, etc.) as well as the data we had to evaluate the model. We run the model with 5 cm resolution and compared it with our original version of 10 cm (see the Figure 1 below presenting the SOC profiles in kg m$^{-3}$ for both resolutions). The results are slightly different mainly because of the thresholds we fixed to define the inputs. For instance, we assume that the aboveground litter goes entirely into the first soil layer. Thus, the concentration in the first layer is higher for finer resolution inducing different diffusive fluxes. We do not fully understand what would be the interest of changing the vertical resolution. The model was not designed to be fully flexible in terms of vertical resolution and it would need an important rewriting of the model. Since the resolution we used correspond to the resolution of the data and it gives good fits we considered that our model outputs are robust enough.

[Figure]

Figure 1. Modeled SOC content (kg m$^{-3}$) with the two pools' model without priming effect with two grid resolutions.

*I can see the point in the current quantifying the priming effect with transferring the decomposition of the priming model to the agroforestry plot with a parameter fixed to conditions of the control (L 627ff). However, I do not agree in the interpretation. To me it is not an absence of priming but is the priming effect as it worked in the control. I still have difficulties in accepting the approach. For me it does not make sense to apply the priming-explicit model with parameter fixed to the control as a no-priming base scenario. Also the results of this run with stock increases of 60t/ha in 18 years are not reasonable (Fig. 6). I am still curious on the comparison of the current quantification to the (to my opinion more straightforward and without much work obtainable) model comparisons I suggested in the previous review involving comparison of priming-explicit vs. no-priming model variants. Therefore, I cannot follow the numbers in abstract of priming effects reducing potential SOC storage by 75 to 90%. (In addition one should differentiate between tree rows/alleys and different depth)*

**Response:** We agree with the reviewer, the reference in the control plot does not correspond to an absence of priming, but to the priming effect as it worked in the control, i.e. induced by the annual crop inputs. This was not clearly stated in the manuscript. However, we were here interested in quantifying the additional priming effect induced by the trees and the tree row vegetation. Applying the priming explicit model with parameter fixed to the control was necessary to have a baseline in the agroforestry plot of the priming effect induced by the crops. It is not possible to quantify the priming effect intensity by just comparing priming-explicit vs. no-priming model variant, because the decomposition rate of the no-priming effect variant implicitly takes priming into account: as a consequence, total modeled SOC stocks were slightly higher with the priming-explicit model variant than with the no-priming model variant (116.68 tC/ha vs 116.28 tC/ha in the tree rows, 105.34 tC/ha vs 106.27 tC/ha in the alleys).

But we agree with the reviewer that this methodology and its interpretation in hard to grasp, and this will probably be the case for future readers. As this part is not central in our study, we decided to remove all this paragraph concerning the estimation of the priming intensity, in the Materials and Methods (P29-30-L612-642) and in the results (P39-L40-L734-749). Figure 6 and figure S3 were then removed. It also has the advantage to reduce the manuscript length.

**Detailed comments**

*Table 5 caption: "the prior values that minimized J(x)". Why is the prior information different for different model variants? I hope that you did not try/optimize for different prior information means. Prior information is defined as an information that is independent of the observations and the calibration. Please add an additional column with mean and standard deviation of the prior parameter pdf.*

**Response:** Here, we used a classical gradient method for reasons that were not clearly explained in the manuscript. We have used a variational scheme to minimize a cost function based on the calculation of the gradient of the cost function with respect to the parameter assuming that all uncertainties are Gaussian (Tarantola, 1987). This is a strong assumption that greatly simplifies the calculation and characterization of the solution.

- First, we should mention that this study is a part of a bigger project aiming to represent the soil C profile in the global land surface model ORCHIDEE (Krinner et al., 2005) with different plant functional type on the same grid. A study case with an agroforestry plot was ideal to test the structure of the model. One objective is to be able to optimize the vertically-discretized soil carbon model embedded in ORCHIDEE using various data streams. In a first step, we decided to start with a simple model, isolated from ORCHIDEE, to be able to run several test without the computing needs of a land surface model. The second step will be to use the whole model and to optimize with a variational approach all parameters of ORCHIDEE. The choice of a variational approach versus a monte carlo one follows from running time constrain that may be prohibitive with a monte carlo approach and a complex process-based model.

- Following that point, we need to recall that we have already performed several parameter optimization studies with the ORCHIDEE model using the same variational approach (Santaren et al 2007, Verbeeck et al. 2011, Kuppel et al. 2012) and that through these studies we gained confidence that the level of non linearities considered in the proposed soil carbon models do not prevent the variational algorithm to obtain a satisfactory solution, provided few cautions. The only requirements or cautions are the need to perform several optimization starting from different prior parameter values, randomly distributed in their allowed range of variation. We then select the case that provides the lowest cost function. With this approach we are much less sensitive to potential local minima. This explain why we have different prior values.

*Fig. 4. I is still interesting to compare depth profiles across control, tree row, and alley. I agree that the closeness of the values makes a combined plot hard to read. But it is even harder to compare across facets. Please, make an attempt to display at least the data or one variant of the prediction in one plot.*

**Response:** We made the following plot to compare depth profiles across control, tree row and alley:

[Figure]

Figure 2. Measured SOC content in the 18-year-old agroforestry system.

As these data were already published in Cardinael et al. (2015) (*Geoderma*), it was not added to the manuscript that already contains many graphs. It could be done if the reviewer thinks it is necessary.

*Fig 5. First I though the [10,11] entry in legend of the bottom subplot was an error of omitting the decimal point, but the change of magnitude and the non-continuous scale are deliberate. Please, make this really obvious and discuss.*

**Response:** Two parallel oblique line were added on the graph to symbolize the change in magnitude in the scale. We also added the following sentence in the legend: "The scale used in the middle and bottom panels are not continuous due to the large stocks predicted by the model in the top layer in the tree-row." (P38-L731-733). It is also discussed P39-L758-764 and P42-43-L820-823.

*Eq 12 (and other eq.): Notations (here d for \partial) dFoc/dt and dFoc_t,z,d/dt are ambiguous. Eq. 14 for dFoc/dt involves terms specific to t,z,d. I suggest renaming all dFoc/dt to dFoc_t,z,d/dt and renaming all previous dFoc_t,z,d/dt to dec_Foc_t,z,d as it only describes decomposition/mineralization. Similar for HSOC.*

**Response:** We agree these equations were ambiguous, we therefore modified them as requested by the reviewer (P12-L261; P14-L295-296; P14-L302-303; P17-L358-372).

*L 649: Suggest modifying "importance of of priming" to "importance of additional priming due to tree row vegetation and trees"*

**Response:** This sentence and this paragraph were removed (see above)

*L 654: I do not agree with "correspond to the absence of priming due to trees". To my opinion it corresponds to "same magnitude of the priming effect as in the control". See may general comments on my difficulties with the current priming effect quantification.*

**Response:** This sentence and this paragraph were removed (see above)

*L 567: Please write more specifically what is "the associated uncertainty". In the previous sentence the known is "These parameters". The 93 soil samples estimate the uncertainty of observed C stocks. How did you use the uncertainty of the observed stocks in the optimization?*

**Response:** Since we used a classical gradient method, the BFGS method needs as input the variance of the observation. We therefore associated a variance matrix read by the algorithm.

*L 605: Manzoni et al 2012 is not a good citation for BIC. Please go back to the original literature cited therein.*

**Response:** This reference has been deleted, and was replaced by Schwarz, 1978 (P28-L596-599).

*Eq. 21: I still do not understand the reasons for using an unusual variant of BIC when you*

*defined your Likelihood by the first term in eq. 21. In effect the current BIC variant uses some different Likelihood that ignores observation uncertainty used in matrix R in eq. 21. It might be an adequate Likelihood for the Manzoni 2012 studied system, but I cannot see how it applies to your case. It would have been very easy to report the requested standard BIC formulation alongside your variant.*

**Response:** We changed the BIC variant to come back to the standard BIC formulation using the likelihood (P28-L596-599). Absolute values have changed, but relative values did not (results are the same for the model comparison) (P37, Fig. 4).

*L 836: Why do the correlations hinder you to compute prediction uncertainty? They are important to consider in Bayesian analysis. The straightforward way would be an MCMC run on eq. 21 to obtain a proper sample from posterior parameter distribution. Perhaps more easy: From the curvature at the optimum you already got and report correlations of a first approximation of a multivariate normal of the parameters posterior distribution. You can draw samples from this distribution and do (say ~1000) forward model runs to compute 95% confidence interval on predictions.*

**Response:** As explained in one of the answer above, we used a BFGS algorithm which is as efficient as a MCMC to reduce the cost function if enough set of parameters are used to launch the algorithm. In our case, we used 30 different sets of priors to avoid local minima, we do believe that the cost function was well minimized as suggested by the Figure 3 below where the cost function reached a limit after the use of 15 set of priors.

[Figure]

Figure 3. Value of the cost function related to the number of prior sets.

*L 902ff: I suggest moving the part on differences in soil temperature and soil moisture and the sensitivity study to the part where you describe what you have not done "simplifying assumptions". Instead of strongly concluding that OC inputs drive SOC storage I recommend formulating more weakly that in this study OC inputs are sufficient to explain the differences in stocks. (You did not compare to explicitly modelling the differences in temperature and moisture with the agroforestry)*

**Response:** The part on differences in soil temperature and soil moisture as well as the sensitivity study was moved above, in the simplifying assumptions (P41-L774-786). We also softened our conclusion as suggested by the reviewer: "This strong validation also revealed that OC inputs were sufficient to explain the differences in SOC stocks at this site." (P41-L789-790).

*L 996: Its a result section. "results were not significantly different" ? Hence, ...*

[revised manuscript text omitted]

---

## Author Response (AR3)

**ASSOCIATE EDITOR**

*I am pleased to announce that both of the two reviewers have recommended publication with minor revisions. Please carefully address the last few technical and methodological comments by the reviewers and provide me with a complete list of your responses to the reviewer comments that either clearly refer to changes in the final revision or explain, where you refrained from changing the manuscript according to the reviewer suggestions.*

*Append a track changes version and a final version of your revised manuscript.*

**Response:** We answered all points raised by the reviewers and made the requested changes.

**REFEREE 1**

*I thank the authors for taking into account my concerns.*

*Regarding the differences between topsoil and tree alleys, the results are now clearer. Thanks for the new Fig. 2 in the replies. This shows nicely that the differences are almost entirely in topsoil and only for the tree row (but also reinforces my concerns for numerical errors in topsoil because of the strong gradients compared to the used resolution). I think this figure is necessary and should be added to the manuscript, because it helps to clarify the results and sets the context for comparing agroforestry with control. The wording in the abstract needs to reflect this too (L 34ff).*

**Response:** This figure was added to the manuscript (P8, Fig. 1). The following sentence was added to the abstract: "However, most of the SOC storage occurred in the first 30 cm of soil and the tree rows." (P2-L34-35).

*Regarding depth resolution, Fig. 1 in reply dampens my concerns on numeric error, as it suggests some robustness. Please, add some discussion to the manuscript on this point in addition to the reply.*

**Response:** The following sentence was added into the discussion: "The strong SOC gradient observed in the topsoil in tree rows compared to the used resolution could lead to numerical errors. We therefore tested a finer grid resolution, and results were very similar, suggesting some robustness of the model." (P39-L749-752).

*Regarding the priming effects, it is kind to omit the interpretation from the manuscript that I was not able to follow.*

**Response:** Thank you. If the reviewer was not able to understand this part, future readers would also have had the same troubles. The best decision was to remove it from the manuscript as it was not the core message.

*Regarding the prior information, I am still confused and have still concerns, see below.*

*You seem not be interested in prior and posterior parameter distributions. To my reading, rather, a prior-like term is used to regularize the cost function. Hence, I suggest to avoid the term Bayesian. The result stay valid also with a gradient based optimization.*

**Response:** When referring to the optimization procedure, the term Bayesian was removed and replaced by "gradient based optimization" (P25-L551-552, P26-L565).

*In the reply on my questions regarding the prior you write about starting parameters for the gradient search instead of prior knowledge on parameters. That is why I am still confused. Starting parameters are a very different thing from priors on parameters. Its good practice to vary the starting values of a gradient search, but its conceptually wrong to randomly vary prior parameter values (aside from testing whether your prior has a significant impact on the posterior.)*

**Response:** To avoid confusion we modified the term "prior" to "starting parameter" in the entire manuscript (P26-L569-571, P26-L575-576, P26-L582, P32 Table 5, P34-L662).

*Did you vary also the prior parameter values in x_b and the P-Matrix (eq. 34) during optimization? To my opinion an optimized prior would be conceptually not acceptable and I would request to redo the optimization with fixed values of x_b and P. This is true for both, a Bayesian analysis and for a regularization of the cost function. Still you need to document values of P (at least diagonals). Or you may omit the prior term in eq. 34 altogether, if you do not need a regularization. If you are not able to redo the optimization, please, clearly describe what you have done with the priors (x_b and P in eq. 34), maybe in an appendix. Although, varying priors is conceptually wrong, you may be confident that it does not have a big effect on results and does not impair conclusions.*

**Response:** No, the starting parameter values were fixed in eq. 34, we clarified this point in the manuscript: '$x_b$ the vector of a starting parameter values fixed for each optimization procedures,' (P26-L575-576). The P-matrix were added in the supplementary materials.

*Please, be precise when speaking about prior parameter values in the cost function (x_b) versus starting values for the optimization.*

**Response:** To avoid confusion we modified the term "prior" to "starting parameter" in the entire manuscript (P26-L569-571, P26-L575-576, P26-L582, P32 Table 5, P34-L662).

*Tab 5: If the optimization is done at original (not log) scale, I suggest to also present the results at original scale.*

**Response:** Table 5 was modified to the previous version of the manuscript (P32).

*L34: The wording suggests that the stocks in the agroforestry plot are increased across all the area and all the depth. While this is true for a total comparison, I strongly suggest rewording so that the facts from new Fig 2 (mainly topsoil, mainly tree alleys) become clearer.*

**Response:** The following sentence was added to the abstract: "However, most of the SOC storage occurred in the first 30 cm of soil and the tree rows." (P2-L34-35).

**REFEREE 2**

*I have reviewed this manuscript once more following the additional revisions. I think the authors have adequately addressed the additional comments, although I am not qualified to judge the issue raised by the other reviewer regarding the use of different priori values in the Bayesian analysis and the use of the BFGS over the suggested MCMC, this evaluation must therefore rely on the other reviewer's assessment. While this study includes some limitations and simplifying assumptions, I think the authors have made sufficient effort in outlining and considering these in the revised manuscript version. Overall this study is a valuable contribution to the scientific literature in my opinion. I have just one minor comment regarding the revised abstract where the deletion of one sentence has created some confusing statement suggesting that 'the priming effect variant being able to capture the SOC depth distribution ...questions the potential of soil to store C' (lines 37-39). Since these two sentences did not refer to each other in the original abstract, the authors should carefully reconsider the wording and link between these remaining sentences in the revised abstract.*

**Response:** Thank you for you kind comments. We rephrased this sentence as requested: "
[revised manuscript text omitted]